# POINT-SAM: PROMPTABLE 3D SEGMENTATION MODEL FOR POINT CLOUDS

**Yuchen Zhou**[1,3] *   **Jiayuan Gu**[2] *   **Tung Yen Chiang**[3]   **Fanbo Xiang**[1]   **Hao Su**[1,3]

[1]Hillbot Inc.   [2]ShanghaiTech University   [3]UC San Diego

## ABSTRACT

The development of 2D foundation models for image segmentation has been significantly advanced by the Segment Anything Model (SAM). However, achieving similar success in 3D models remains a challenge due to issues such as non-unified data formats, poor model scalability, and the scarcity of labeled data with diverse masks. To this end, we propose a 3D promptable segmentation model **Point-SAM**, focusing on point clouds. We employ an *efficient* transformer-based architecture tailored for point clouds, extending SAM to the 3D domain. We then distill the rich knowledge from 2D SAM for Point-SAM training by introducing a data engine to generate part-level and object-level pseudo-labels at scale from 2D SAM. Our model outperforms state-of-the-art 3D segmentation models on several indoor and outdoor benchmarks and demonstrates a variety of applications, such as interactive 3D annotation and zero-shot 3D instance proposal.

## 1 INTRODUCTION

The development of 2D foundation models for image segmentation has been significantly advanced by *Segment Anything* (Kirillov et al., 2023). That pioneering work includes a promptable segmentation task, a segmentation model (SAM), and a data engine for collecting a dataset (SA-1B) with over 1 billion masks. SAM shows impressive zero-shot transferability to new image distributions and tasks. Thus, it has been widely used in many applications, e.g., segmenting foreground objects for image-conditioned 3D generation (Liu et al., 2023c;a), NeRF (Cen et al., 2023b), and robotic tasks (Wang et al., 2023; Chen et al., 2023).

*Can we just lift SAM to create 3D foundation models for segmentation?* Despite a few efforts (Yang et al., 2023; Xu et al., 2023; Zhou et al., 2023b) to extend SAM to the 3D domain, those existing approaches are limited to applying SAM on 2D images and then lifting the results to 3D. This process is constrained by image quality, and thus is likely to fail for textureless or colorless shapes like CAD models (Lambourne et al., 2021). Besides, it is also affected by view selection. Too few views may not adequately cover the entire shape, while too many views can significantly increase the computational burden. Moreover, it can suffer from multi-view inconsistency when results are merged from different views, since they may conflict and be impacted by occlusions. Furthermore, multi-view images only capture surface, making it infeasible to label internal structures, essential for annotating articulated objects (e.g. drawers in a cabinet). Therefore, it is necessary to develop native 3D foundation models to address the aforementioned limitations.

However, developing native 3D foundation models, or extending SAM to the 3D domain, presents several challenges: 1) **There is no unified representation for 3D shapes.** 3D shapes can be represented by meshes, voxels, point clouds, implicit functions, or multi-view images, while 2D images are usually represented by a dense grid of pixels. Unlike 2D images, 3D shapes can vary significantly in scale and sparsity. For example, indoor and outdoor datasets often cover different ranges and typically require different models. 2) **There are no unified network architectures in the 3D domain.** Due to the heterogeneity of 3D data, different network architectures have been proposed for different representations, such as PointNet (Qi et al., 2017a) for point clouds and SparseConv (Graham et al., 2018) for voxels. 3) **It is more difficult to scale up 3D networks.** 3D networks are natively more computationally costly. For instance, SAM utilizes deconvolution and

---

*Equal Contributions. Corresponding Authors: yuz256@ucsd.edu, gujy1@shanghaitech.edu.cn

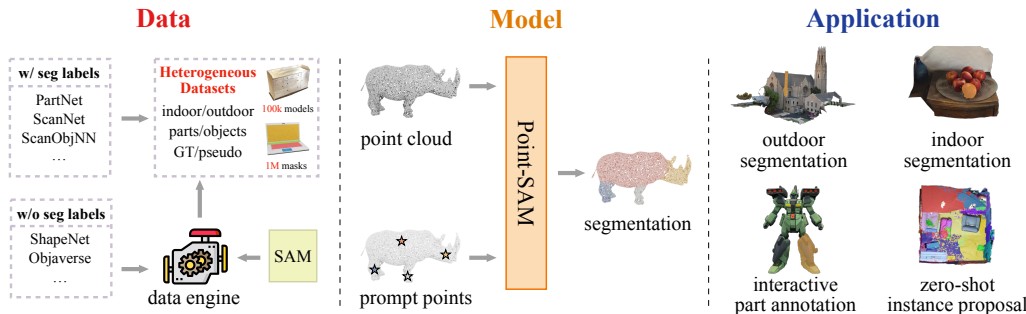

Figure 1: We propose a 3D extension of SAM, named **Point-SAM** (Sec. 3), which predicts masks given the input point cloud and prompts. To scale up training data, we develop a data engine (Sec. 4) to generate pseudo labels with the help of SAM. The final models, trained on a mixture of datasets, are capable of handling data from various sources and producing results at multiple levels of granularity. We demonstrate the versatility and efficacy of our approach through multiple applications and downstream tasks, as detailed in Sec. 5.

bilinear upsampling in its decoder, whereas there are no 3D operators for point clouds as efficient as their 2D counterparts. 4) **High-quality 3D labels, especially those with diverse masks, are rare.** SAM is initially trained on existing datasets with ground-truth labels of low diversity, and then used to facilitate annotating more masks at different granularity (e.g., part, object, semantics) to increase label diversity. However, in the 3D domain, existing datasets contain only a small number of segmentation labels. For example, the largest dataset with part-level annotations, PartNet (Mo et al., 2019), includes only about 26,671 shapes and 573,585 part instances.

In this work, our goal is to build a 3D promptable segmentation model for point clouds, as a foundational step towards 3D foundation models. Point clouds are selected as our primary representation, because other representations can be readily converted into point clouds, and real-world data is often captured in this format. Following SAM, we address 3 critical components: **task**, **model**, and **data**. We focus on the **3D promptable segmentation task**, which involves predicting valid segmentation masks in response to any given segmentation prompt. To address this task, we propose a 3D extension of SAM, named **Point-SAM**. We utilize a transformer-based encoder to embed the input point cloud, alongside a point prompt encoder for point prompts, and a mask prompt encoder for mask prompts. Point-cloud and prompt embeddings are fed to a transformer-based mask decoder to predict segmentation masks. To efficiently encode point clouds and pointwise prompts, we develop a novel tokenizer based on Voronoi diagram to obtain point-cloud embeddings, as input to the transformer-based encoder. Regarding data, we train Point-SAM on **a mixture of heterogeneous datasets**, including PartNet and ScanNet (Dai et al., 2017), with both part- and object-level annotations. To expand label diversity and leverage large-scale unlabeled datasets such as ShapeNet (Chang et al., 2015), we have developed a data engine to generate pseudo labels with the assistance of SAM. This pipeline enables us to distill knowledge from SAM, and our experiments demonstrate that these pseudo labels significantly improve zero-shot transferability. Our contributions include:

- We develop a 3D promptable segmentation model **Point-SAM**, adept at processing point clouds from various sources in a unified way. A novel tokenizer based on Voronoi diagram is proposed to efficiently embed point clouds and dense prompts.
- We propose a data engine to generate pseudo labels with substantial mask diversity by distilling knowledge from SAM. It is shown to significantly enhance our model's performance on out-of-distribution (OOD) data.
- Our experiments demonstrate the strong zero-shot transferability of our model to unseen point cloud distributions and new tasks, positioning it as a 3D foundation model.

## 2 RELATED WORK

**Lifting 2D foundation models for 3D segmentation** Despite the growing number of 3D datasets, high-quality 3D segmentation labels remain scarce. To address this, 2D foundation models trained on web-scale 2D data, such as CLIP (Radford et al., 2021), GLIP (Li et al., 2022), and SAM (Kirillov

et al., 2023), have been leveraged. A prevalent framework involves adapting these 2D foundation models for 3D applications by merging results across multiple views (Liu et al., 2023d) (Zhou et al., 2024) (Cen et al., 2024). SAM3D (Yang et al., 2023) and SAMPro3D (Xu et al., 2023) utilize RGB-D images with known camera poses to lift SAM to segment 3D indoor scenes. PartSLIP (Liu et al., 2023b; Zhou et al., 2023b), dedicated to part-level segmentation, first renders multiple views of a dense point cloud, then employs GLIP and SAM to segment parts, and finally consolidates multi-view results into 3D predictions. These methods are limited by the capabilities of 2D foundation models and the quality of multi-view rendering. Besides, they usually require complicated and slow post-processing to integrate multi-view results, which also poses challenges in maintaining multi-view consistency. Another strategy involves distilling knowledge from 2D foundation models directly into 3D models. For example, Segment3D (Huang et al., 2023) and SAL (Ošep et al., 2024) both utilize SAM to generate pseudo labels given RGB images and train native 3D models on scene-level point clouds. However, these approaches can only handle surface points, making it difficult to segment internal structures that are common in part-level segmentation of articulated 3D shapes such as cabinets with drawers.

**3D foundation models** The development of 3D foundation models has advanced notably. Point-BERT (Yu et al., 2022b) proposes a self-supervised paradigm for pretraining 3D representations for point clouds. OpenShape (Liu et al., 2024) and Uni3D (Zhou et al., 2023a) scale up 3D representations with multi-modal contrastive learning. (Hong et al., 2023) trains 3D-based Large Language models (3D-LLM) on collected diverse 3D-language data, utilizing 2D pretrained VLMs. LEO (Huang et al., 2024), sharing similar ideas, focuses on embodied ability such as navigation and robotic manipulation. Our work concentrates on 3D segmentation. Despite several initiatives aimed at open-world 3D segmentation, such as OpenScene (Peng et al., 2023) and OpenMask3D (Takmaz et al., 2024), these primarily address scene-level segmentation and are trained on relatively small datasets.

**3D interactive segmentation** Interactive segmentation has been explored across both 2D and 3D domains. (Kirillov et al., 2023) introduces a groundbreaking project including the promptable segmentation task, the 2D foundation model (SAM), and a data engine to collect large-scale labels. In the 3D domain, InterObject3D (Kontogianni et al., 2023) and AGILE3D (Yue et al., 2023) share similar ideas to segment point clouds while their training is confined to ScanNet (Dai et al., 2017). In contrast, our model is designed to handle both object- and part-level segmentation, leveraging a wide range of datasets including CAD models and real scans. Thus, our model shows greater versatility and adaptability. Besides, 3D interactive segmentation is also explored within implicit representations. SA3D (Cen et al., 2023b) enables users to achieve 3D segmentation of any target object through a single one-shot manual prompt in a rendered view. SAGA (Cen et al., 2023a) distills SAM features into 3D Gaussian point features through contrastive training. While these methods necessitate an additional optimization process, our model operates on a feed-forward basis and can respond within seconds, offering a more efficient solution.

## 3 POINT-SAM

In this section, we present Point-SAM, a promptable segmentation model for point clouds. Fig. 2 provides an overview of Point-SAM. Inspired by SAM (Kirillov et al., 2023), Point-SAM consists of 3 components: a point-cloud encoder, a prompt encoder, and a mask decoder. Unlike 2D models, Point-SAM addresses unique challenges related to point clouds: computation efficiency, scalability, and irregularity. We denote the input point cloud as $P \in \mathbb{R}^{N \times 3}$ and its point-wise feature as $F \in \mathbb{R}^{N \times \bar{D}}$.

**Point-cloud encoder with Voronoi tokenizer** The point-cloud encoder transforms the input point cloud into a point-cloud embedding. Inspired by recent advancements in 3D point-cloud transformers (Zhao et al., 2021; Wu et al., 2022; Yu et al., 2022a; Zhou et al., 2023a), we employ a similar transformer-based encoder. Concretely, it first selects a fixed number of centers $C \in \mathbb{R}^{L \times 3}$ using farthest point sampling (FPS), and groups the k-nearest neighbors of each center as a *patch*. A local PointNet (Qi et al., 2017a) is used to tokenize each patch $G_{patch} \in \mathbb{R}^{L \times K \times (3+\bar{D})}$. The resulting patch tokens $F_{patch} \in \mathbb{R}^{L \times D}$, combined with the positional embeddings of the group centers, are processed by a Vision Transformer (Dosovitskiy et al., 2021) to generate the final point-cloud embedding $F_{pc} \in \mathbb{R}^{L \times D}$.

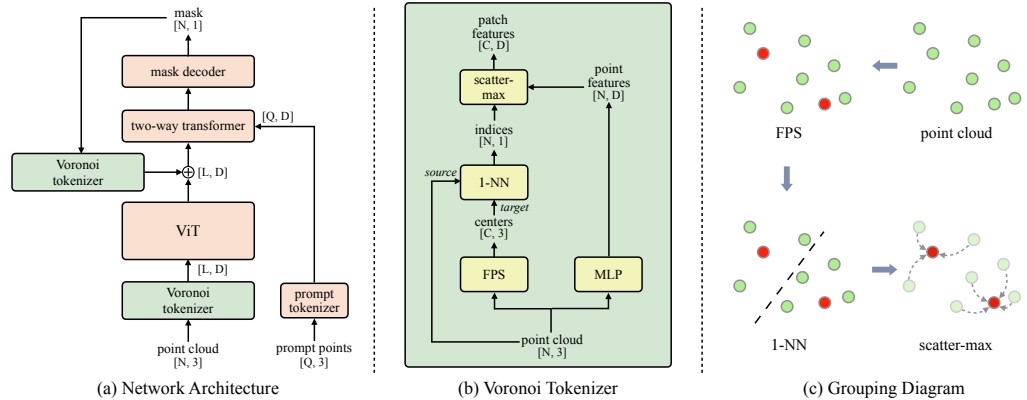

(a) Network Architecture     (b) Voronoi Tokenizer     (c) Grouping Diagram

Figure 2: Overview of Point-SAM. (a) illustrates the overall network architecture. The model takes a point cloud along with several point prompts as inputs. Initially, the point cloud is divided into patch tokens using a **Voronoi tokenizer**. After that, the patch tokens are embedded through a vanilla Vision Transformer (ViT). The token features are then fused with the mask features from the previous iteration. The a two-way transformer is employed to allow interaction with the features of the prompt points. Finally, a lightweight decoder generates the mask output. (b) depicts the design of the Voronoi tokenizer, where a **Voronoi diagram** is used for grouping the high-resolution point cloud into patch tokens, instead of relying on traditional K-nearest neighbors (KNN) methods. (c) provides a visual diagram of the grouping process within the Voronoi tokenizer.

We observe that $L \times K$ is typically much larger than $N$, making the process of tokenizing point clouds to obtain $F_{patch}$ time-consuming and memory-intensive. To address this, we propose a novel tokenizer based on the **Voronoi diagram**, which strikes a balance between efficiency and effectiveness. Specifically, we group points by assigning each point to its nearest center, forming a Voronoi diagram where each patch corresponds to a Voronoi cell. An MLP is then used to extract pointwise features $\hat{F}_{patch} \in \mathbb{R}^{N \times D}$ based on the relative position of each point to its nearest center. Patch tokens $F_{patch} \in \mathbb{R}^{L \times D}$ are max-pooled within each Voronoi cell via a scatter-max operator.

**Prompt encoder** The prompt encoder encodes various types of prompts into prompt embeddings. In this work, we focus on two types of prompts: points and masks. Point prompts are processed similarly to SAM. Each point is associated with a binary label indicating whether it is a foreground prompt. These prompts are encoded to their positional encodings (Tancik et al., 2020) $F_{point} \in \mathbb{R}^{Q \times D}$, summed with learned embeddings indicating their labels. $Q$ denotes the number of point prompts. Mask prompts are represented as pointwise logits $X_{mask} \in \mathbb{R}^{N \times 1}$, typically derived from the model's previous predictions. These logits are concatenated with the input point cloud's coordinates and processed through a tokenizer described in the point-cloud encoder. The resulting mask prompt embeddings $F_{mask} \in \mathbb{R}^{L \times D}$ are element-wise summed with the point-cloud embedding.

**Mask decoder** The mask decoder efficiently maps the point-cloud embedding, prompt embeddings, and an output token $F_{out} \in \mathbb{R}^{1 \times D}$ into a segmentation mask $Y_{mask} \in \mathbb{R}^{N \times 1}$. We follow SAM to employ two Transformer decoder blocks that use prompt self-attention and cross-attention in two directions (prompt-to-point-cloud and vice versa), to update all embeddings. Different from the 2D counterpart, we upsample the updated point-cloud embedding $F_{pc} \in \mathbb{R}^{L \times D}$ to match the input resolution by using inverse distance weighted average interpolation based on 3 nearest neighbors (Qi et al., 2017b), followed by an MLP. The upsampled point-cloud embedding is denoted as $X_{pc} \in \mathbb{R}^{N \times D}$. Another MLP transforms the output token to the weight of a dynamic linear classifier $X_{out} \in \mathbb{R}^{1 \times D}$, which calculates the mask's foreground probability at each point location as $Y_{mask} = X_{pc} \cdot X_{out}^T$. Consistent with SAM, our model can generate multiple output masks for a single point prompt by introducing multiple output tokens. Note that multi-mask outputs are enabled only when there is only a single point prompt with no mask prompts. In addition, we also introduce another token $F_{iou} \in \mathbb{R}^{M \times D}$ to predict the IoU score for each mask output, where $M$ is the number of multiple mask outputs.

Table 1: Summary of training datasets.

| Dataset | PartNet | PartNet-Mobility | ScanNet | ScanNet-block | Fusion360 | ShapeNet | Overall |
|---------|---------|------------------|---------|---------------|-----------|----------|---------|
| Number of point clouds | 16442 | 2163 | 1198 | 24328 | 35000 | 20000 | 99131 |
| Average number of masks | 13 | 16 | 30 | 12 | 2 | 17 | 10 |
| Has ground truth labels | True | True | True | True | True | False | - |

**Training**  Mask predictions are supervised with a weighted combination of focal loss (Lin et al., 2017) and dice loss (Milletari et al., 2016), in line with SAM. We simulate an interactive setup, detailed in Sec. 5.1, by sampling prompts across 7 iterations per mask. The loss for mask prediction is computed between the ground truth mask and the predictions at all iterations. More details are provided in App. A. For multiple mask outputs, we follow SAM to use a "hindsight" loss, where we only back-propagate only the minimum loss over masks. Additionally, the predicted IoU score is supervised using a mean squared error loss. For training, we randomly sample 10,000 points as input. Besides, we normalize the input point to fit within a unit sphere centered at zero, to standardize the inputs. The number of patches $L$ and the patch size $K$ are set to 512 and 64 by default.

**Inference with variability**  A significant challenge in handling 3D point clouds is their irregular input structure; the number of points can vary, necessitating a dynamic approach to group points into a varying number of patches with adjustable sizes. While previous point-based methods (Zhou et al., 2023a) are typically limited to processing a fixed number of points, our model's flexible design allows it to handle larger point sets than those used during training, by adjusting the number of patches and the patch size. Unless otherwise specified, we set the number of patches and the patch size to 2048 and 512 when the number of input points exceeds 32768. In contrast, voxelization-based methods (Yue et al., 2023) struggle with such variations as changing voxel resolution can significantly impact performance, the results with different voxel resolutions are shown in App. B.

## 4  TRAINING DATASETS

**Integrating existing datasets**  Foundation models are typically data-hungry, and the diversity of segmentation masks is crucial to support "segment anything". Thus, we use a mixture of existing datasets with ground truth segmentation labels, which are summarized in Table 1. We utilize synthetic datasets including the training split of PartNet (Mo et al., 2019), PartNet-Mobility (Xiang et al., 2020), and Fusion360 (Lambourne et al., 2021). Since PartNet does not provide textured meshes, we only keep the models that are from ShapeNet where textured meshes are available. We use all part hierarchies of PartNet. For PartNet-Mobility, we hold out 3 categories (scissors, refrigerators, and doors) not included in ShapeNet, which are used for evaluation on unseen categories. For PartNet and Fusion360, we uniformly sample 32768 points from mesh faces. For each object in PartNet-Mobility, we render 12 views, fuse point clouds from rendered RGB-D images, and sample 32768 points from the fused point cloud using Farthest Point Sampling (FPS). For scene-level datasets, we use the training split of ScanNet200 (Dai et al., 2017) and augment it by splitting each scene into blocks. The augmented version is denoted as ScanNet-Block. Concretely, we use a 3m×3m block with a stride of 1.5m. We use FPS to sample 32768 points per scene or block.

**Generating pseudo labels**  Existing datasets lack sufficient diversity in masks. Large-scale 3D datasets like ShapeNet (Chang et al., 2015) usually do not include part-level segmentation labels. Besides, most segmentation datasets only provide exclusive labels, where each point belongs to a single instance. To this end, we develop a data engine to generate pseudo labels.

Figure 3 illustrates the pseudo label generation process. Initially, Point-SAM is trained on the mixture of existing datasets. Next, we utilize both pre-trained Point-SAM and SAM to generate pseudo labels. Concretely, for each mesh, we render RGB-D images at 6 fixed camera positions and fuse a colored point cloud. SAM is applied to generate diverse 2D proposals for each view. For each 2D proposal, we intend to find a 3D proposal corresponding to it. We start from the view corresponding to the 2D proposal. A 2D prompt is randomly sampled from the 2D proposal and lifted to a 3D prompt, which prompts Point-SAM to predict a 3D mask on the fused point cloud. Then, we sample the next

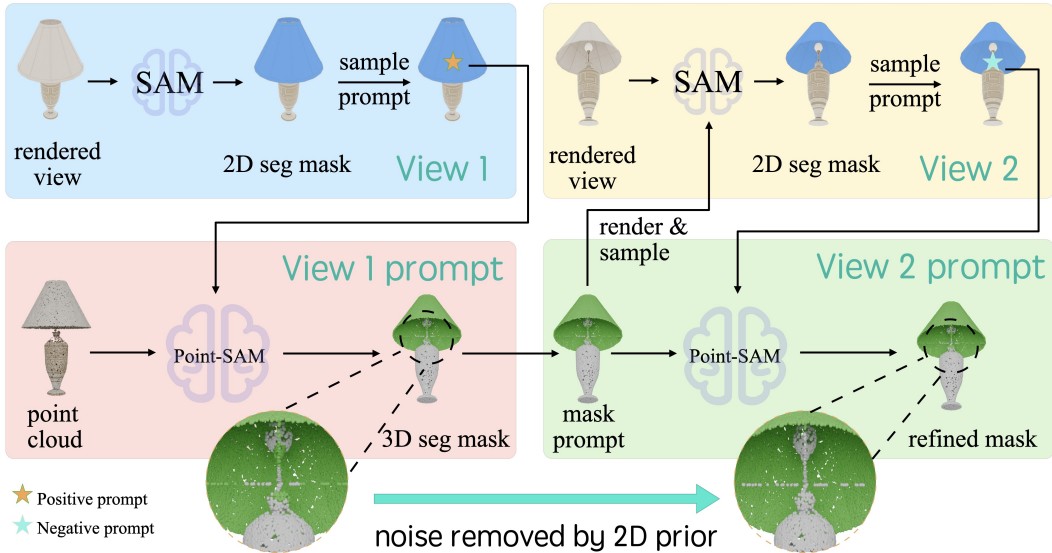

Figure 3: Illustration of pseudo label generation. Initially, we select one segmentation mask from the instance proposals ("segment everything") generated by SAM on the first view. Then, we prompt Point-SAM by lifting 2D prompt points to 3D (**View 1 prompt**). Subsequently, the 3D segmentation mask output by Point-SAM is refined using additional views. We first prompt SAM by projecting the 3D segmentation mask onto the second view (**View 2**), leveraging SAM's strong prior knowledge to revise the mask. Then, we sample more 2D prompt points from the revised area by SAM, and prompt Point-SAM again by lifting these points to 3D (**View 2 prompt**).

2D prompt from the error region between the 2D proposal and the projection of the 3D proposal at this view. New 3D prompts and previous 3D proposal masks are fed to Point-SAM to update the 3D proposal. The process is repeated until the IoU between the 2D proposal and the projection of the 3D proposal is larger than a threshold. This step ensures 3D-consistent segmentation regularized by Point-SAM while retaining the diversity of SAM's predictions. We repeat the above process with a few modifications at other views to refine the 3D proposal. At other view, we first sample the initial 2D prompt from the projection of previous 3D proposal, which is used to prompt SAM to generate multiple outputs. The output 2D mask with the highest IoU relative to the projection is selected as the "2D proposal" in the previous process. If the IoU is lower than a threshold, the 3D proposal is discarded. Previous 3D proposal mask is used to prompt Point-SAM at each iteration. This step aids in refining the 3D masks by incorporating 2D priors from SAM through space carving. We use our data engine to generate pseudo labels for 20000 shapes from ShapeNet. On average, each shape is annotated with 17 masks, offering a diversity comparable to PartNet.

## 5 EXPERIMENTS

We have conducted the experiments showing the strong zero-shot transferability and the superior efficiency of our method. Experiments are conducted on zero-shot point-prompted segmentation (Sec. 5.1), few-shot part segmentation (App. B.2) and zero-shot object proposal generation (App. B.1). Furthermore, we showcase an application of 3D interactive annotation in our supplementary materials.

### 5.1 ZERO-SHOT POINT-PROMPTED SEGMENTATION

**Task and metric**    The task is to segment instances based on 3D point prompts. For automatic evaluation, point prompts need to be selected. We adopt the same method to simulate user clicks described in Kontogianni et al. (2023). In brief, the first point prompt is selected as the "center" of the ground truth mask, which is the point farthest from the boundary. Each subsequent point is chosen from two candidates: one from the false-positive set at the farthest minimum distance to the complementary set, and the other from the false-negative set selected similarly. Then, the candidate farther from the boundary is selected. See App. C for details. This evaluation protocol is commonly

Table 2: Summary of evaluation datasets.

| Property | PartNet-Mobility | ScanObjectNN | S3DIS | KITTI360 | Replica | Overall |
|---|---|---|---|---|---|---|
| Number of point clouds | 125 | 2,723 | 68 | 379 | 17 | 3312 |
| Average number of masks | 6 | 3 | 33 | 9 | 152 | 5 |
| Average number of points | 10,000 | 157,669 | 522,058 | 40,000 | 1,308,124 | - |

used in prior 2D (Kirillov et al., 2023; Zhang et al., 2024) and 3D (Kontogianni et al., 2023; Yue et al., 2023) works on interactive single-object segmentation. Following those prior works, we use the metric **IoU@k**, which is the Intersection over Union (IoU) between ground truth masks and prediction given $k$ point prompts. The metric is averaged across instances.

**Datasets** We evaluate on a heterogeneous collection of datasets, covering both indoor and outdoor data, along with part- and object-level labels. For part-level evaluation, we use the synthetic dataset PartNet-Mobility (Xiang et al., 2020) and the real-world dataset ScanObjectNN (Uy et al., 2019). As mentioned in Sec. 4, we hold out 3 categories of PartNet-Mobility for evaluation. In the same way as the training dataset, we render 12 views for each shape, fuse a point cloud from multi-view depth images, and sample 10,000 points for evaluation. ScanObjectNN contains 2902 objects of 15 categories collected from SceneNN (Hua et al., 2016) and ScanNet (Dai et al., 2017). For scene-level evaluation, we use S3DIS (Armeni et al., 2016) and KITTI-360 (Liao et al., 2022). Specifically, we use the processed data from AGILE3D (Yue et al., 2023), which contains scans cropped around each instance. Table 2 summarizes the datasets used for evaluation.

**Baselines** We compare Point-SAM with a multi-view extension of SAM, named **MV-SAM**, and a 3D interactive segmentation method, **AGILE3D** (Yue et al., 2023). Inspired by previous works (Yang et al., 2023; Xu et al., 2023; Zhou et al., 2023b) that lift SAM's multi-view results to 3D, we introduce MV-SAM for zero-shot point-prompted segmentation as a strong baseline. First, we render multi-view RGB-D images from the mesh of each shape. Note that mesh rendering is needed to ensure high-quality images, which are essential for good SAM's performance. Thus, this baseline actually has access to more information than ours. Then we prompt SAM at each view with the simulated click sampled from the "center" of the error region between the SAM's prediction and 2D ground truth mask. The predictions are subsequently lifted back to the sparse point cloud (10,000 points) and merged into a single mask. If a point is visible from multiple views, its foreground probability is averaged. For both MV-SAM and our method, we select the most confident prediction if there are multiple outputs. AGILE3D is close to our approach, while it uses a sparse convolutional U-Net as its backbone and is only trained on the real-world scans of ScanNet40. Besides, it does not normalize its input, and thus it is sensitive to object scales. To process CAD models without known physical scales, we adjust the scale of the input point cloud for AGILE3D, so that its axis-aligned bounding box has a maximum size of 5m, determined through a grid search (App. B.3).

**Results** Table 3 presents the quantitative results. Point-SAM shows superior zero-shot transferability and effectively handle data with different numbers of points as well as from different sources. Point-SAM significantly outperforms MV-SAM, especially when only few point prompts are provided, while MV-SAM achieves reasonably good performance with a sufficient number of prompts. Notably, for **IoU@k**, MV-SAM actually samples $k$ prompts per view. It indicates that our 3D native method is more prompt-efficient. Besides, it is challenging for SAM to achieve multi-view consistency without extra fine-tuning, especially with limited prompts. Moreover, Point-SAM also surpasses AGILE3D across all datasets, particularly in out-of-distribution (OOD) scenarios such as PartNet-Mobility (held-out categories) and KITTI360. It underscores the strong zero-shot transferability of our method and the importance of scaling datasets. Figure 4 shows the qualitative comparison between Point-SAM, AGILE3D and MV-SAM, where Point-SAM demonstrates superior quality with a single prompt and significantly faster convergence compared to AGILE3D and MV-SAM.

Table 3 also compares the Voronoi tokenizer with the previous KNN tokenizer. We observe that the Voronoi tokenizer achieves comparable performance to the KNN tokenizer, while showing superior efficiency. We test the time and memory efficiency on a single Nvidia RTX-4090 GPU using point clouds from KITTI360. For each point cloud, 10 prompt points are sampled. The Voronoi tokenizer increases the frames per second (FPS) by **163.4%** ($5.2 \rightarrow 13.7$ shape/sec) and reduces GPU memory usage by **18.4%** ($3890 \rightarrow 3172$ MB).

Table 3: Quantitative results for zero-shot point-prompted segmentation. The notation **Voronoi** indicates the use of the Voronoi tokenizer and **KNN** indicates the use of the KNN tokenizer, while all other settings remain unchanged.

| Dataset | Method | IoU@1 | IoU@3 | IoU@5 | IoU@7 | IoU@10 |
|---------|--------|-------|-------|-------|-------|--------|
| PartNet-Mobility | AGILE3D | 26.4 | 40.8 | 50.8 | 57.4 | 61.9 |
| | MV-SAM | 29.3 | 57.0 | 69.7 | 74.3 | 76.9 |
| | Ours (Voronoi) | 46.0 | **68.1** | 73.2 | 76.0 | 77.8 |
| | Ours (KNN) | **47.9** | 67.7 | **74.2** | **77.0** | **78.6** |
| ScanObjectNN | AGILE3D | 34.8 | 52.0 | 61.6 | 67.2 | 72.3 |
| | Ours (Voronoi) | **52.4** | **76.3** | 81.4 | 84.0 | 85.7 |
| | Ours (KNN) | 49.4 | 75.3 | **82.0** | **84.8** | **86.3** |
| S3DIS | InterObject3D++ | 32.7 | 69.0 | 80.8 | - | 89.2 |
| | AGILE3D | 58.7 | 77.4 | 83.6 | 86.4 | 88.5 |
| | Ours (Voronoi) | **63.6** | **81.8** | 84.8 | 86.4 | 87.3 |
| | Ours (KNN) | 47.6 | 78.4 | **86.2** | **89.2** | **90.4** |
| KITTI360 | InterObject3D++ | 3.4 | 11.0 | 19.9 | - | 40.6 |
| | AGILE3D | 34.8 | 42.7 | 44.4 | 45.8 | 49.6 |
| | Ours (Voronoi) | **52.8** | 71.4 | 81.3 | 83.9 | 85.7 |
| | Ours (KNN) | 49.4 | **74.4** | **81.7** | **84.3** | **85.8** |

Table 4: Ablation study on training dataset. We report the IoU@k metrics for zero-shot prompt-segmentation on PartNet-Mobility (held-out categories).

| Training dataset | PartNet | PartNet +ScanNet | PartNet +ShapeNet | PartNet +ShapeNet+ScanNet | Full |
|------------------|---------|------------------|-------------------|---------------------------|------|
| IoU@1 | 38.2 | 39.7 | 44.5 | 45.4 | 47.9 |
| IoU@3 | 56.3 | 58.6 | 65.2 | 66.5 | 67.7 |
| IoU@5 | 60.6 | 68.8 | 71.8 | 72.6 | 74.2 |
| IoU@10 | 63.5 | 71.9 | 76.2 | 77.5 | 78.6 |

## 5.2 ABLATIONS

**Scaling up datasets** Previous works have been limited by the size and scope of their training datasets. For example, AGILE3D (Yue et al., 2023) was trained solely on ScanNet (Dai et al., 2017), which includes only 1,201 scenes. As detailed in Table 1, our training dataset encompasses 100,000 point clouds, 100 times larger than ScanNet. To verify the effectiveness of scaling up training data, we conduct an ablation study on dataset size and composition. We introduce 4 dataset variants: 1) PartNet only, 2) PartNet+ScanNet (including ScanNet-Block), 3) PartNet+ShapeNet (pseudo labels), and 4) PartNet+ShapeNet+ScanNet. We train Point-SAM on these variants, resulting in different models. Table 4 shows the comparison of these models in zero-shot prompt segmentation on PartNet-Mobility (held-out categories). The model trained on PartNet+ScanNet surpasses the one trained solely on PartNet, although the evaluation dataset (part-level labels) has a markedly different distribution from the added ScanNet (object-level labels). Moreover, the model trained on PartNet+ShapeNet achieves even better performance, particularly with a single prompt. Note that the IoU@1 metric assesses whether the model captures sufficient mask diversity, since a single prompt is inherently ambiguous and the ground-truth label depends on dataset bias. It suggests that our pseudo labels effectively incorporate part-level knowledge distilled from SAM. Furthermore, it is observed that the zero-shot performance on out-of-distribution data consistently improves, as we utilize increasingly larger and more diverse data.

**Trained on ScanNet only** To ensure a fair comparison with AGILE3D, we conduct an ablation study, where Point-SAM is trained exclusively on ScanNet, following the same settings as AGILE3D. This resulting model is denoted as Point-SAM*. Table 5a presents the comparison among AGILE3D, Point-SAM* and the original Point-SAM. Provided similar training data, Point-SAM* significantly

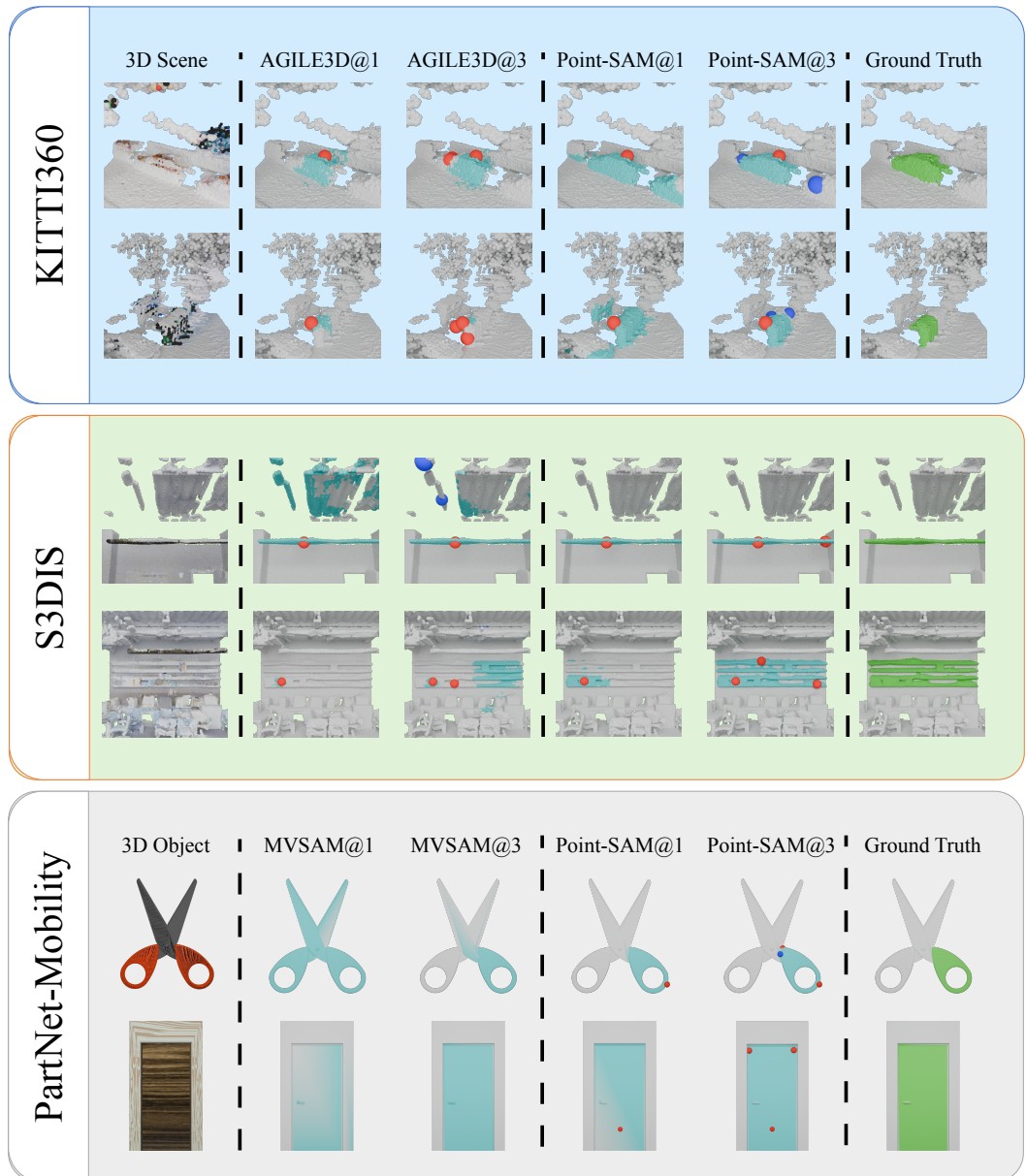

Figure 4: Qualitative results of prompt segmentation are presented for three different settings: KITTI360 for zero-shot outdoor scene segmentation, S3DIS for indoor scene segmentation, and PartNet-Mobility for zero-shot part segmentation. We compare our results with AGILE3D on KITTI360 and S3DIS, and with MVSAM on PartNet-Mobility. Point-SAM demonstrates superior segmentation results with fewer prompt points across all three datasets. Red points represent positive prompt points, while blue points indicate negative prompt points.

outperforms AGILE3D on the OOD datasets like KITTI360 and PartNet-Mobility, although performing worse on the in-domain dataset S3DIS. We hypothesize that AGILE3D is highly optimized for ScanNet (the only dataset it was trained on), and some of its design choices may lead to overfitting to this dataset, like its objective of simultaneously generating exclusive multiple objects.

**Data Engine**  To demonstrate the effectiveness of distilling knowledge from SAM, we conduct an ablation study on our design of the data engine. We compare our current pipeline with a baseline that uses a pre-trained Point-SAM to generate instance proposals as pseudo labels. Specifically, we sample

Table 5: Ablation studies on model design and data engine.

(a) Quantitative results for Point-SAM trained on ScanNet. Point-SAM* is trained on ScanNet following the setting of AGILE3D. Point-SAM refers to our original setting.

| Dataset | Method | IoU@1 | IoU@3 | IoU@5 | IoU@7 | IoU@10 |
|---|---|---|---|---|---|---|
| PartNet-Mobility | AGILE3D | 26.4 | 40.8 | 50.8 | 57.4 | 61.9 |
| | Point-SAM* | 33.5 | 48.5 | 57.0 | 61.2 | 66.8 |
| | Point-SAM | **47.9** | **67.7** | **74.2** | **77.0** | **78.6** |
| ScanObjNN | AGILE3D | 34.8 | 52.0 | 61.6 | 67.2 | 72.3 |
| | Point-SAM* | 32.0 | 56.7 | 63.2 | 68.8 | 70.5 |
| | Point-SAM | **49.4** | **75.3** | **82.0** | **84.8** | **86.3** |
| S3DIS | AGILE3D | **58.7** | 77.4 | 83.6 | 86.4 | 88.5 |
| | Point-SAM* | 38.8 | 67.1 | 72.2 | 78.9 | 80.6 |
| | Point-SAM | 47.6 | **78.4** | **86.2** | **89.2** | **90.4** |
| KITTI360 | AGILE3D | 34.8 | 42.7 | 44.4 | 45.8 | 49.6 |
| | Point-SAM* | 44.0 | 67.1 | 72.2 | 78.9 | 80.8 |
| | Point-SAM | **49.4** | **74.4** | **87.1** | **84.3** | **85.8** |

(b) Ablation study on the data engine. Different models are trained on different pseudo label datasets, and evaluated on PartNet-Mobility. * means using pseudo labels generated by Point-SAM ("segment everything") without SAM.

| PartNet Mobility | PartNet | PartNet +ShapeNet | PartNet +ShapeNet* |
|---|---|---|---|
| IoU@1 | 38.2 | 39.6 | 44.5 |
| IoU@3 | 56.3 | 65.2 | 56.7 |
| IoU@5 | 60.6 | 71.8 | 64.6 |
| IoU@10 | 63.5 | 76.2 | 67.8 |

Table 6: Sensitivity to point count. We report the IoU@k metrics for zero-shot prompt-segmentation on S3DIS, with varying the number of patches and patch size.

| (#patches, patch size) | (512,64) | (512,256) | (2048,64) | (2048,256) |
|---|---|---|---|---|
| IoU@1 | 41.9 | 49.0 | 47.4 | 47.6 |
| IoU@3 | 64.2 | 69.9 | 74.3 | 78.4 |
| IoU@5 | 72.2 | 76.4 | 81.7 | 86.2 |
| IoU@10 | 76.7 | 80.2 | 85.8 | 90.5 |

1,024 prompt points from the point cloud, using each point to prompt Point-SAM. Non-Maximum Suppression (NMS) is applied to filter out duplicate instances. Table 5b compares the models trained on pseudo labels generated by our pipeline and the baseline. Interestingly, Point-SAM even benefits from pseudo labels generated by itself. Moreover, incorporating 2D SAM plays a crucial role in improving the quality of the pseudo labels, leading to a substantial boost in overall performance.

**Sensitivity to Point Count**    Point clouds are typically irregular. When handling point clouds with more points than those used in our training, we have to adjust the number of patches and the patch size accordingly. Thus, we conduct experiments to study the effect of these two hyperparameters. Table 6 shows the qualitative results of zero-shot prompt-segmentation on S3DIS (Armeni et al., 2016). We select S3DIS, because the average number of points for S3DIS is about 500K, 50 times larger than that of our training datasets. Our results indicate that it is important to increase the number of patches to accommodate larger point clouds. Enlarging the patch size is also crucial due to the different neighborhood densities compared to our training distribution.

## 6    CONCLUSION

In conclusion, our work presents significant strides towards developing a foundation model for 3D promptable segmentation using point clouds. By adopting a transformer-based architecture, we have successfully implemented Point-SAM, which effectively and efficiently responds to 3D point and mask prompts. Our model leverages a robust training strategy across mixed datasets like PartNet and ScanNet, which has proven beneficial, especially when enhanced with pseudo labels generated through our novel pipeline that distills knowledge from SAM.

However, there are inherent limitations and challenges in our approach. The diversity and scale of the 3D datasets used still lag behind those available in 2D, posing a challenge for training models that can generalize well across different 3D environments and tasks. Furthermore, the computational demands of processing large-scale 3D data and the complexity of developing efficient 3D-specific operations remain significant hurdles. Our reliance on pseudo labels, while beneficial for expanding label diversity, also introduces dependencies on the quality and variability of the 2D labels provided by SAM, which may not always capture the complex nuances of 3D structures.

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

## A   TRAINING DETAILS

**Training recipe.** Point-SAM is trained with the AdamW optimizer. We train Point-SAM for 100k iterations. The learning rate (*lr*) is set to 5e-5 after learning rate warmup. Initially, the *lr* is warmed up for 3k iterations, starting at 5e-8. A step-wise *lr* scheduler with a decay factor of 0.1 is then used, with *lr* reductions at 60k and 90k iterations. The weight decay is set to 0.1. The training batch size for Point-SAM, utilizing ViT-g as the encoder, is set to 4 per GPU with a gradient accumulation of 4, and it is trained on 8 NVIDIA H100 GPUs with a total batch size of 128. The ViT-l version can be trained across 2 NVIDIA A100 GPUs, with a batch size of 16 per GPU and gradient accumulation of 4, for 50k iterations. For Point-SAM utilizing ViT-l as the backbone, the step-wise learning rate decay milestones are set at 30k and 40k iterations.

**Data augmentation.** We apply several data augmentation techniques during training. For each object, we pre-sample 32,768 points before training and then perform online random sampling of 10,000 points from these 32,768 points for actual training. We apply a random scale for the normalized points with a scale factor of [0.8, 1.0] and a random rotation along y-axis from $-180°$ to $180°$. For object point clouds we also apply a random rotation perturbation to x- and z-axis. The perturbation angles are sampled from a normal distribution with a standard deviation (sigma) of 0.06, and then these angles are clipped to the range [0, 0.18].

## B   ADDITIONAL EXPERIMENTS

### B.1   ZERO-SHOT OBJECT PROPOSALS

In this section, we evaluate Point-SAM on zero-shot object proposal generation. The ability to automatically generating masks for all possible instances is known as "segment everything" in SAM. SAM samples a 64x64 point grid on the image as prompts, and uses Non-Maximum Suppression (NMS) based on bounding boxes to remove duplicate instances. We adapt this approach for 3D point clouds with some modifications. First, we sample prompts using FPS, and then prompt Point-SAM to generate 3 masks per prompt. For post-processing, a modified version of NMS based on point-wise masks is applied.

We compare with OpenMask3D (Takmaz et al., 2024) on Replica (Straub et al., 2019). OpenMask3D utilizes a class-agnostic version of Mask3D (Schult et al., 2023) trained on ScanNet200 to generate object proposals. For our Point-SAM, we sample 1024 prompts and set the NMS threshold to 0.3. In addition, to handle the extensive point counts in Replica, we downsample each scene to 100,000 points and later propagate the predictions to their nearest neighbors at the original resolution. We also adjust the number of patches and the patch size to 4096 and 64 respectively. For both methods, we truncate the proposals to the top 250.

We use the average recall (AR) metric. We filter out "undefined" and "floor" categories from the ground truth labels. Table 7a shows the quantitative results. Point-SAM showcases strong performance compared to OpenMask3D, which is tailored for this task, even though our model is never trained on such a large number of points and is zero-shot evaluated on unseen data. It highlights the robust zero-shot capabilities of our method.

Table 7: Qualitative results of zero-shot object proposal generation and few-shot part segmentation.

(a) Zero-shot object proposal generation on Replica.

| Method | $AR_{25}$ | $AR_{50}$ |
|---|---|---|
| OpenMask3D | 40.2 | **31.5** |
| Ours | **49.2** | **31.5** |

(b) Few-shot part segmentation on ShapeNetPart. The numbers with * are reported by Uni3D (Zhou et al., 2023a).

| | PointBERT | Uni3D(close) | Uni3D(open) | Ours |
|---|---|---|---|---|
| 1-shot | 66.2* | 71.5 | 75.9* | 73.9 |
| 2-shot | 71.9* | 73.8 | 78.2* | 76.1 |

### B.2   FEW-SHOT PART SEGMENTATION

Foundation models can be effectively fine-tuned for various tasks. In this section, we demonstrate that Point-SAM has captured good representations for part segmentation. We compare with

PointBERT (Yu et al., 2022b) and Uni3D (Zhou et al., 2023a) on close-vocabulary, few-shot part segmentation. We use ShapeNetPart (Yi et al., 2016) and report the $mIoU_C$, which is the mean IoU averaged across categories. Similar to Uni3D, we adapt Point-SAM for close-vocabulary part segmentation. Specifically, we extract features from the 4th, 8th, and last layers of the ViT in our encoder and use feature propagation (Qi et al., 2017b) to upscale them into point-wise features, followed by an MLP to predict point-wise multi-class logits. During few-shot training, we freeze our encoder and only optimize the feature propagation layer as well as the MLP using cross-entropy loss. Unlike PointBERT and our method, Uni3D originally aligns point-wise features with text features of ground truth part labels extracted by CLIP. We refer to it as Uni3D (open), since it is designed for open-vocabulary part segmentation. We also evaluate its variant sharing our modification for close-vocabulary part segmentation, denoted as Uni3D (close). Table 7b presents the results for both 1-shot and 2-shot settings. Point-SAM surpasses both PointBERT and Uni3D (close), which indicates that our approach has acquired versatile knowledge applicable to downstream tasks.

### B.3 NORMALIZATION SCALE FOR AGILE3D

We conduct a grid search to determine the optimal normalization scale for AGILE3D on PartNet-Mobility. Table 8 shows the effect of normalization scale for AGILE3D in zero-shot prompt segmentation on PartNet-Mobility (held-out categories). We find that AGILE3D achieves its best performance with a normalization scale of 5.

Table 8: The effect of normalization scale for AGILE3D.

| Normalization scale | IoU@3 | IoU@5 | IoU@7 | IoU@9 |
|---|---|---|---|---|
| 1 | 29.0 | 36.4 | 41.0 | 44.7 |
| 3 | 35.7 | 43.1 | 48.4 | 51.7 |
| 5 | **40.8** | **50.8** | **57.4** | **61.9** |
| 7 | 36.8 | 45.6 | 51.3 | 57.5 |
| 10 | 35.1 | 42.2 | 48.6 | 53.1 |

## C PROMPT SAMPLING

Following AGILE3D (Yue et al., 2023), we sample two prompt points for each instance: one from false positive points and another from false negative points. The prompts are selected by identifying the foreground point that has the furthest distance to the nearest background point. Specifically, this involves computing pairwise distances from foreground points to background points, determining the minimum distance to background points for each foreground point, and selecting the foreground point with the maximum of these minimum distances.

After computing the distances and selecting the candidates, we have two prompt point candidates: the point sampled from false positive points serves as a negative prompt, and the point sampled from false negative points serves as a positive prompt. We select the one with the furthest distance to the nearest background points as the final prompt point.

## D VORONOI TOKENIZER

We shows the whole table of the Voronoi tokenizer efficiency improvement in Table 9. Voronoi-based tokenizer surpasses the KNN-based tokenizer on all the datasets for both time and memory. Different with 2D SAM, the mask encoder of Point-SAM also need a point cloud tokenizer, which makes the voronoi tokenizer more important.

## E TRAINING DATASETS ABLATION

Table 10 presents the ablation study on training datasets evaluated across different datasets. This experiment highlights the scaling-up effect as more data is incorporated. The results demonstrate

Table 9: This table shows the evaluation results of the Voronoi tokenizer. As the Voronoi tokenizer increases the time and memory efficiency significantly, it's very important for the real-world applications.

| | Method | FPS@1 | FPS@5 | FPS@10 | Memory |
|---|---|---|---|---|---|
| PartNet-Mobility | KNN | 22.3 | 11.7 | 7.3 | 2524 |
| | Voronoi | 28.6 (+34.4%) | 22.4 (+82.0%) | 17.8 (+143.8%) | 2078 (-21.5%) |
| ScanObjectNN | KNN | 5.9 | 3.0 | 1.4 | 8548 |
| | Voronoi | 7.3 (+23.7%) | 4.9 (+63.3%) | 2.3 (+62.0%) | 6316 (-26.1%) |
| S3DIS | KNN | 3.31 | 1.57 | 0.93 | 13680 |
| | Voronoi | 3.52 (+34.4%) | 1.72 (+8.7%) | 1.04 (+11.8%) | 10788 (-20.5%) |
| KITTI360 | KNN | 15.7 | 8.9 | 5.2 | 3890 |
| | Voronoi | 21.1 (+34.4%) | 16.2 (+82.0%) | 13.7 (+163.4%) | 3172 (-18.4%) |

Table 10: This table shows the datasets ablation results evaluated on all the evaluation datasets.

| Eval Dataset | Point Number | PartNet | ScanNet | PartNet +ScanNet | PartNet +ShapeNet | PartNet +ScanNet+ShapeNet | Full |
|---|---|---|---|---|---|---|---|
| PartNet -Mobility | IoU@1 | 38.2 | 33.5 | 39.7 | 44.5 | 45.4 | 47.9 |
| | IoU@3 | 56.3 | 48.5 | 58.6 | 65.2 | 66.5 | 67.7 |
| | IoU@5 | 60.6 | 57.0 | 68.8 | 71.8 | 72.6 | 74.2 |
| | IoU@10 | 63.5 | 66.8 | 71.9 | 76.2 | 77.5 | 78.6 |
| ScanObjectNN | IoU@1 | 38.4 | 32.0 | 44.8 | 45.2 | 48.2 | 49.4 |
| | IoU@3 | 63.9 | 56.7 | 71.8 | 72.4 | 73.9 | 75.3 |
| | IoU@5 | 66.8 | 63.2 | 78.9 | 80.2 | 81.4 | 82.0 |
| | IoU@10 | 67.7 | 70.5 | 80.8 | 81.4 | 84.8 | 86.3 |
| S3DIS | IoU@1 | 25.6 | 38.8 | 43.5 | 30.5 | 46.2 | 47.6 |
| | IoU@3 | 48.7 | 65.3 | 73.6 | 54.1 | 75.9 | 78.4 |
| | IoU@5 | 63.2 | 71.4 | 83.9 | 67.9 | 85.8 | 86.2 |
| | IoU@10 | 67.9 | 80.6 | 88.3 | 76.3 | 90.1 | 90.4 |
| KITTI360 | IoU@1 | 36.2 | 44.0 | 46.3 | 39.0 | 47.3 | 49.4 |
| | IoU@3 | 62.9 | 67.1 | 70.5 | 67.8 | 72.1 | 74.4 |
| | IoU@5 | 68.9 | 72.2 | 77.9 | 73.5 | 80.4 | 81.7 |
| | IoU@10 | 74.8 | 80.8 | 83.6 | 80.9 | 85.0 | 85.8 |

that adding data consistently enhances performance, although the extent of improvement varies depending on the type of data and the evaluation dataset. For example, adding ShapeNet leads to greater improvements on PartNet-Mobility, while adding ScanNet has a more pronounced effect on S3DIS. Furthermore, we observe that increasing the dataset size also improves performance on KITTI360, suggesting that the transferability of Point-SAM increases as the amount of training data grows.

# F   MORE VISUALIZATION

We provide additional qualitative results in the appendix. Figure  5 presents qualitative results on the Waymo Open dataset. As a fully out-of-distribution experiment, this figure highlights the transferability of Point-SAM, demonstrating its ability to correctly segment outdoor objects such as cars and trees.

Figure  4 shows the ground truth segmentation results alongside outcomes with varying numbers of prompt points. As Waymo is an OOD dataset, this figure demonstrates the superior transferibility of Point-SAM.

Figure  6 illustrates the few-shot segmentation results on the ShapeNet-Part dataset. The pre-trained model is used as the embedding for linear probing. We randomly select one sample from each category

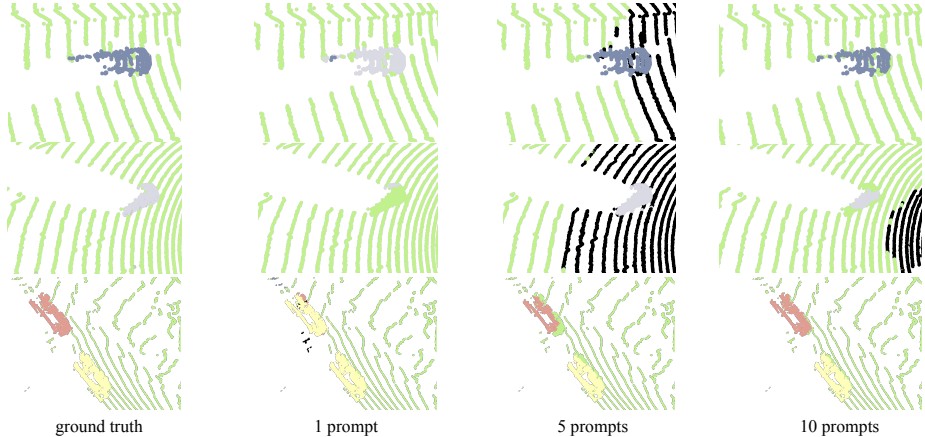

Figure 5: This figure shows the segmentation results of Waymo.

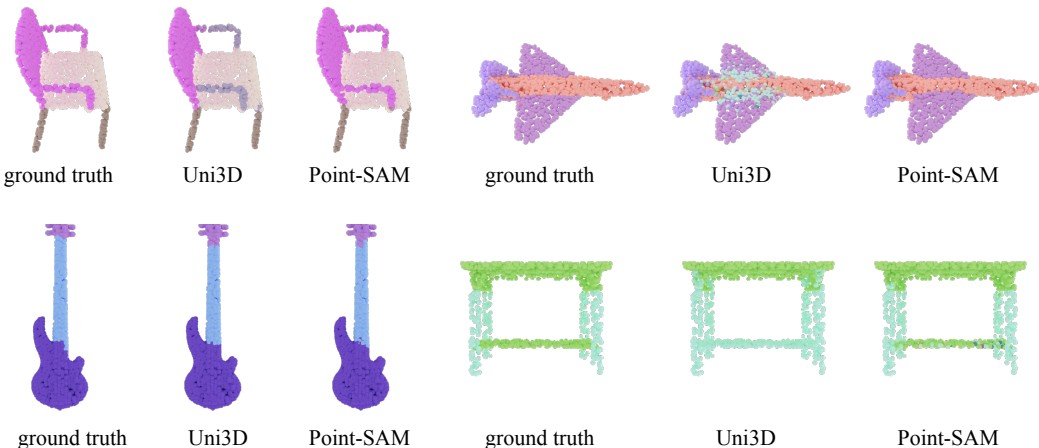

Figure 6: This figure shows the few-shot segmentation results on ShapeNet-Part.

as the training set, then perform inference on the evaluation set to obtain the one-shot probing results. These results demonstrate the superior pre-training quality of Point-SAM for segmentation tasks.

## G   EVALUATION FOR INTERIOR SEGMENTATION

Table 11 shows the segmentation results for the StorageFurniture from PartNet-Moblity. We use the Point-SAM trained only on PartNet, and ScanNet. As shown in Table 11, MV-SAM achieves worse performance on StorageFurniture than the other 3 categories shown in Table 4. Point-SAM achieves better performance than MV-SAM with interior segmentation masks.

Table 11: We present the evaluation results for the StorageFurniture category from PartNet-Mobility. To ensure the evaluation data is not in the training dataset, we used Point-SAM trained on PartNet, ShapeNet and ScanNet.

|          | IoU@1 | IoU@3 | IoU@5 | IoU@7 | IoU@10 |
|----------|-------|-------|-------|-------|--------|
| MV-SAM   | 26.9  | 51.0  | 63.3  | 65.3  | 67.3   |
| Point-SAM| **42.3** | **63.4** | **68.6** | **71.8** | **74.5** |

Table 12: This table presents the evaluation results on Replica. Following the evaluation procedure of AGILE3D , we crop the point clouds centered on the segmentation mask.

|          | IoU@1 | IoU@3 | IoU@5 | IoU@7 | IoU@10 |
|----------|-------|-------|-------|-------|--------|
| AGILE3D  | 55.9  | 74.8  | 81.7  | 85.4  | 87.9   |
| Point-SAM | **58.3** | **79.5** | **86.2** | **90.1** | **91.4** |

## H EVALUATION ON REPLICA

We present the quantitative results on Replica in Table 12. We still follow the progress of evaluating S3DIS and KITTI360 to split the scene into blocks centering on the segmentation target. In this experiment, we use the Point-SAM trained on the whole training dataset with Voronoi-based tokenizer. Point-SAM outperforms AGILE3D on the evaluation of Replica.

## I MORE INTERACTIVE SEGMENTATION RESULTS

Figure 7 shows more visualization results for complicated scenes and objects. We show both the segmentation results projected to meshes and the raw point cloud results. Figure 7 shows the superior transferibility of Point-SAM.

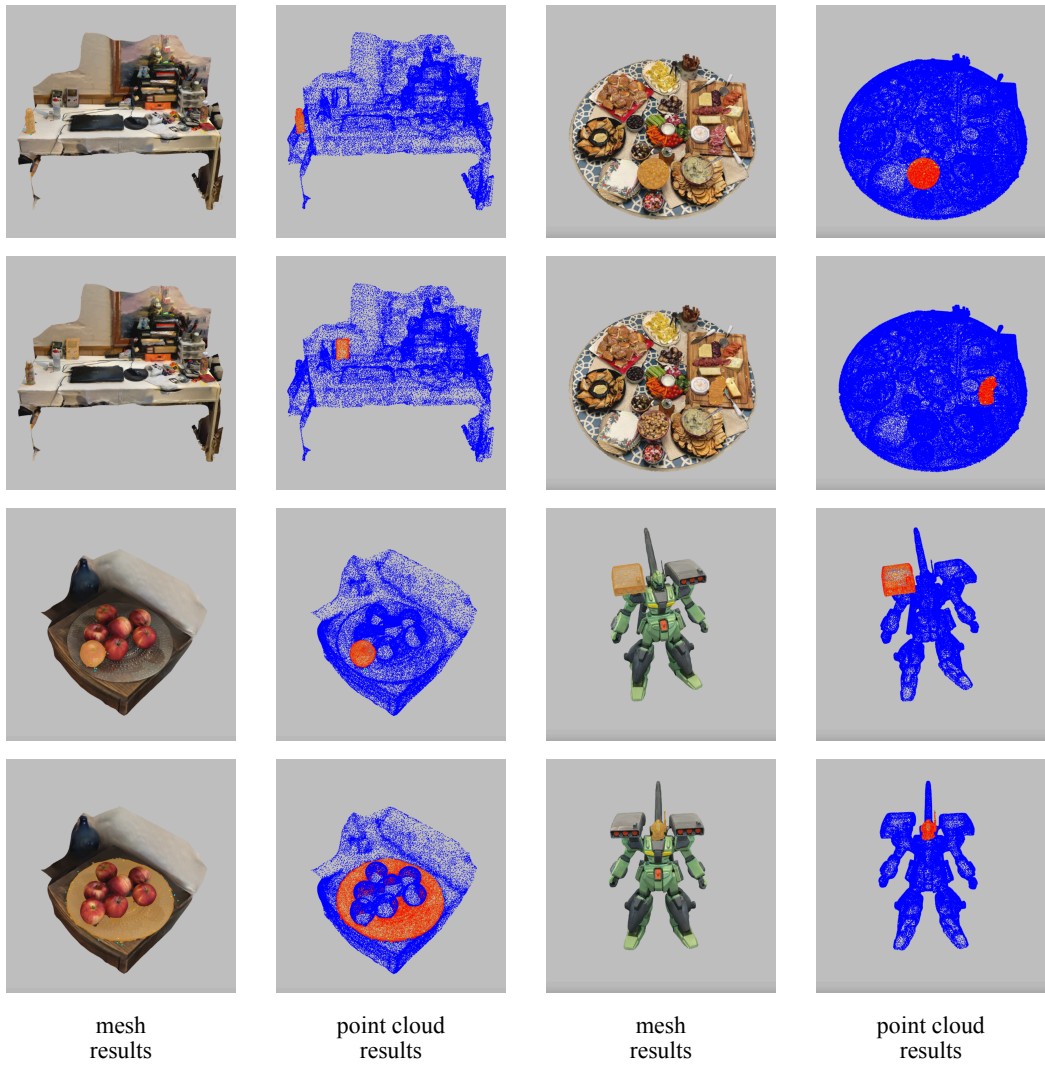

|   mesh
results   |   point cloud
results   |   mesh
results   |   point cloud
results   |

Figure 7: This figure presents additional visualization results of the interactive promptable segmentation. All objects were sourced from Polycam and Objaverse. We show both the projection results on meshes and the raw results on point clouds.

