# OpenReview forum: "Point-SAM: Promptable 3D Segmentation Model for Point Clouds"
_ICLR.cc/2025/Conference — ICLR 2025 Poster_

### Official Review · Reviewer_e1ZC · 2024-11-02

**Soundness:** 2
**Presentation:** 3
**Contribution:** 2
**Rating:** 6
**Confidence:** 5

**Summary:**

This paper is mainly built to tackle 3D promptable segmentation task. The authors propose a segmentation model called Point-SAM, which can take 3D point prompts and mask prompts as input prompt, alongside a Voronoi tokenizer is used to fuse dense prompts. This model is trained on two phases: 1. existing well-labeled datasets; 2. unlabeled datasets, whose pseudo labels are generated by pre-trained Point-SAM from phase 1 and SAM, in order to distilling knowledge from SAM. They compare their method with baselines on zero-shot segmentation task, and showing much better performances.

**Strengths:**

Unlike many 2D-to-3D segmentation lifting work, this paper chooses to distill knowledge from SAM in training phase instead of aggregating in very evaluation, making it more efficient and not requiring running SAM for several times during inference. The shows great performance on choosing datasets, showing great performance on zero-show segmentation task due to distilling and better model design.

**Weaknesses:**

The main contribution of this paper can be divided into two folds: better model design and novel data engine to distill SAM instead of aggregating. So the paper should demonstrate the effectivity for both these two parts. However, the experiments are not strong enough:
1. This model choose to compare with AGILE3D, a straightforward baseline. In main experiments, the experiment is not fair, since the model proposed actually trained on much larger datasets than the baseline. In order to have a fair comparison, they also trained the model with the same training set as baseline in Table 6.a. However, this method only show apparent better performance on 2 out of 4 datasets. The authors argue it may due to that the baseline is overfitting to these kinds of data, but it’s assertion is not strong enough, because these two datasets both tend to include scenes with much more points than the other two. At least a cross analysis ablation, i.e. train A test B and train B test A.
2. The baselines are not thorough and strong enough. It is strange why the baseline is not trained on the same level of data. And the multi-view SAM aggregation baseline is not strong enough. The authors make up one, MV-SAM. However, there are plenty of such multi-view aggregating baselines, such as SAM-guided Graph Cut, SAI3D, Open3DIS, OpenIns3D.
3. In Table 4, evaluating only on PartNet-Mobility is not sufficient. From the table, we observe that adding ScanNet is less beneficial than adding ShapeNet. Further explanation or additional experiments are needed to clarify this difference.

**Questions:**

Besides the questions in weakness, I also have some other questions:
1. The authors listed Replica as an evaluation datasets in Table 2, but only compared with OpenMask3D in appendix but not with other baselines. Why do you make this choice? Combined with what mentioned in Weakness 1 and Table 6, is this due to your model is very sensitive and not so well-defined for large scale data?
2. According to Figure 4, for KITTI360 and S3DIS datasets, are the red and blue balls the visualization of prompt points? If so, why the prompts to Point-SAM and AGILE3D are not consistent? Do you use the same prompt in your main experiments?
I am open to raising my score if the authors can thoroughly address the weaknesses and questions.

---

> ### Author Response · Authors · 2024-11-23
>
> > Concerns about comparison with AGILE3D
> >
>
> Thank you for your suggestions. As shown in Table 6a, Point-SAM, when trained on the same dataset as AGILE3D, outperforms AGILE3D on KITTI360 and PartNet-Mobility, achieves comparable performance on ScanObjectNN, but performs worse on S3DIS. We want to point out that AGILE3D benefits from the comparison on S3DIS, as AGILE3D is specifically designed and tuned for indoor scene segmentation, making S3DIS very similar to the distribution of the training dataset.
>
> In this paper, we are not claiming a better backbone significantly outperforms AGILE3D in a fair comparison. Our main contribution is to propose a general method that works across different data types and sources. Therefore, we believe that Table 4 is sufficient to demonstrate the reasonableness of our model design. Additionally, Table 3 showcases the superior performance of our entire pipeline compared to the baselines, providing further evidence that our method is the first 3D promptable segmentation approach that works effectively across a wide range of datasets.
>
> > Baseline not strong enough
> >
>
> The reason we designed MV-SAM as our multi-view aggregation baseline is that most previous works in this area focus on language-guided 3D instance proposals, including the four works you mentioned. To the best of our knowledge, there is no straightforward multi-view aggregation baseline specifically designed for promptable 3D segmentation. We would greatly appreciate any suggestions for a multi-view aggregation baseline that works with point prompts and should be considered for comparison.
>
> We found that promptable segmentation is a particularly challenging task for aggregation-based methods because the 3D prompt points are invisible from some views, making it difficult for SAM to generate 2D segmentation results. To address this, we designed a stronger baseline, MV-SAM, which samples prompt points on each view according to the error regions. This design benefits MV-SAM in the comparison, as more prompt points are sampled. We believe this approach makes MV-SAM a sufficiently strong baseline for our task.
>
> > Lack of ablation datasets
> >
>
> Thank you for your suggestion. We have added Table 10 in our revision, which presents the results of the dataset ablation evaluated across all datasets. Table 10 shows that adding ShapeNet provides a greater performance boost on PartNet-Mobility, while adding ScanNet has a more significant impact on S3DIS. As expected, incorporating similar datasets tends to be more beneficial. However, Table 10 also demonstrates that adding dissimilar datasets can still improve performance, highlighting the model's ability to generalize across diverse data sources.
>
> > No evaluation on Replica
> >
>
> Thank you for pointing out the lack of Replica evaluation. Replica is not included in AGILE3D's evaluation datasets, and given its high similarity to S3DIS, we initially excluded it from the promptable segmentation experiment. We have included an additional experiment in Table 12 of the revised manuscript. This experiment demonstrates that Point-SAM, trained on the entire training datasets with the Voronoi tokenizer, outperforms AGILE3D.
>
> > Confusion about Figure 4
> >
>
> Thank you for highlighting the absence of legends for Figure 4, we have added in the revision. The red balls represent positive prompt points, while the blue ones indicate negative prompt points. The selection of prompt points follows the same sampling rule as AGILE3D, which is also used in previous 2D promptable segmentation methods. The first prompt point is chosen at the center of the ground truth mask. For subsequent iterations, two prompt candidates are selected from the centers of false positive and false negative points. The candidate further from the region’s border is then chosen as the prompt point.
>
> In our qualitative comparison with baseline methods, we apply the same rule for selecting prompt points for both our method and the baselines. In this context, the first prompt point is the same for both methods, but the subsequent prompt points vary based on each method's output. Both the position and the signal of these points depend on the method's output and the selection rule, leading to differences between our approach and the baselines. Since both methods follow the same sampling rule, the comparison is fair.

---

> > ### Comment · Reviewer_e1ZC · 2024-11-26
> > **Reviewer's Response**
> >
> > Thanks to the authors for their responses, which addressed most of my concerns. As a result, I am willing to raise my rating to 6.

---

> > > ### Author Response · Authors · 2024-11-26
> > >
> > > We sincerely thank you for the invaluable feedback and positive assessment. Your expertise and the rating upgrade are truly appreciated.

---

> > > ### Author Response · Authors · 2024-11-30
> > >
> > > We sincerely thank you for taking the time to review our work and for your positive assessment. We humbly noticed that the rating has not yet been updated. As the discussion deadline is approaching, we would be deeply grateful if you could kindly consider modifying the rating at your earliest convenience. Thank you so much for your understanding and support.

---

### Official Review · Reviewer_y8Cn · 2024-11-03

**Soundness:** 3
**Presentation:** 3
**Contribution:** 2
**Rating:** 6
**Confidence:** 5

**Summary:**

This paper presents a method for achieving generalizable 3D point segmentation by leveraging prior knowledge from the 2D Segment Anything framework.

**Strengths:**

Propose a better tokenizer and show reasonable performance improvement compared with baseline.

**Weaknesses:**

Missing references:

"Segment Anything 3D"
"SANeRF-HQ: Segment Anything for NeRF in High Quality"
"SERF: Fine-Grained Interactive 3D Segmentation and Editing with Radiance Fields"
"Segment Anything in 3D with Radiance Fields"

**Questions:**

Could you clarify the difference between the proposed pseudo label generation method and "SERF: Fine-Grained Interactive 3D Segmentation and Editing with Radiance Fields"?

Additionally, the proposed Voronoi approach shows only minimal improvement compared to KNN. Could you clarify the efficiency comparison between these two?

But overall, it is a good work.

---

> ### Author Response · Authors · 2024-11-23
>
> > Missing references
> >
>
> Thanks for your suggestion. We have added the citations of these papers in our revision.
>
> > Comparison with SERF
> >
>
> Thank you for pointing out this prior work. The key difference between our pseudo-label generation method and SERF lies in leveraging prior knowledge from existing 3D segmentation datasets.
>
> The most challenging aspect of pseudo-label generation is obtaining segmentation masks for other views that accurately correspond to the segmentation mask from the first view. Both SERF and our method tackle this challenge by generating prompt points and using them to prompt SAM for other views. SERF selects prompt points by identifying the closest visible point to the previous prompt points, relying on basic 3D spatial relationships and the precision of the previous prompt points.
>
> In contrast, our approach uses a pre-trained Point-SAM model trained on existing 3D segmentation datasets to bridge the connection between views. For any mask on the first view, our pre-trained Point-SAM generates clearer and more precise segmentation results on other views, providing a stronger initialization for segmentation in new views. With this better initialization, we can select more reasonable prompt point on new views and prompt SAM to give us better segmentation masks.
>
> We believe the prior knowledge learned from human-annotated segmentations enhances the quality of pseudo-labels, aligning them more closely with human semantics.
>
> > Voronoi tokenizer improvement
> >
>
> We have included a more detailed Table 9 in our revision, from L813 to L824, which highlights the significant performance improvements achieved by the Voronoi tokenizer across all test datasets. While its performance is comparable to the KNN tokenizer, the Voronoi tokenizer provides a substantial boost in inference speed—over 80% faster on average—while also reducing GPU memory usage by more than 15%. As demonstrated by SAM, inference speed is critical for real-world applications, making the Voronoi tokenizer particularly valuable.

---

> ### Author Response · Authors · 2024-11-28
>
> We sincerely appreciate the time and effort you have dedicated to reviewing our work and providing invaluable feedback. We kindly ask if our rebuttal has adequately addressed all of your concerns. If so, we would greatly appreciate it if you could consider adjusting the rating accordingly.

---

### Official Review · Reviewer_nzz5 · 2024-11-04

**Soundness:** 3
**Presentation:** 3
**Contribution:** 3
**Rating:** 6
**Confidence:** 4

**Summary:**

The paper adheres to the philosophy of the Segment Anything Model by proposing Point-SAM, a 3D promptable segmentation model specifically designed for point clouds. It primarily makes two contributions: (1) a scalable and efficient transformer-based architecture that facilitates 3D promptable segmentation; and (2) a data engine that distills knowledge from the Segment Anything Model to generate pseudo labels. Experiments conducted on various indoor and outdoor datasets have demonstrated its zero-shot transferability and significant application potential.

**Strengths:**

- The authors develop a data engine to generate pseudo labels, providing sufficient diverse masks for datasets lacking ground truth. This allows Point-SAM to be trained with more heterogeneous datasets.
- Following the design philosophy of SAM, the implementation on 3D point clouds is shown through experiments to endow Point-SAM with zero-shot capabilities.
- The Voronoi tokenizer achieves comparable performance to the KNN tokenizer, while showing superior efficiency in KITTI360 dataset.

**Weaknesses:**

- The challenges presented in the introduction are not well sovled by the proposed method. For instance, the paper mentions "There is no unified representation for 3D shapes," yet does not provide a unified representation and only designs for point cloud representation.
- While the authors have made efforts to train Point-SAM on a diverse set of datasets, the reliance on synthetic datasets and the dominance of indoor scenes may limit the model's generalizability to outdoor and more varied environments. The paper would benefit from an evaluation on a broader range of datasets, particularly those capturing diverse outdoor scenes such as Waymo Open Dataset [1] and nuScenes [2].
- Although the paper claims efficiency improvements with the Voronoi tokenizer and tests it on KITTI360 with 10 prompt points, a comprehensive analysis of the computational costs is lacking. The authors should provide more detailed benchmarks comparing Point-SAM with other methods in terms of both speed and memory usage.

[1] Sun, Pei, et al. "Scalability in perception for autonomous driving: Waymo open dataset." Proceedings of the IEEE/CVF conference on computer vision and pattern recognition. 2020.

[2] Caesar, Holger, et al. "nuscenes: A multimodal dataset for autonomous driving." Proceedings of the IEEE/CVF conference on computer vision and pattern recognition. 2020.

**Questions:**

Please see the weaknesses section.

---

> ### Author Response · Authors · 2024-11-23
>
> > Confusion about unified representation
> >
>
> Thank you for pointing out the confusion in the presentation. As explained in L77-79, the intent of this sentence is to highlight that 3D data can be represented in various ways, making it challenging to create a model that works across all representations. Therefore, we chose point clouds as the input to our model, as transferring from other representations to point clouds is feasible and straightforward
>
> > Outdoor scenes evaluation
> >
>
> Thank you for your suggestion. We have added the Waymo evaluation results in Figure 5 of the revision, from L865 to L876. As shown in Figure 5, we present the segmentation results for all instances, using 1, 5, and 10 prompt points for each instance. These three results are displayed separately, alongside the corresponding ground truth labels for comparison. While the segmentation results with 1 prompt point are not working well, the results significantly improve with 5 and 10 prompt points. This demonstrates that our model learns the promptable segmentation process rather than overfitting to the data prior.
>
> > Voronoi tokenizer evaluation
> >
>
> Thank you for your suggestion. We have included a more detailed table in our paper, from L813 to L824, as well as in the general response which highlights the significant performance improvements achieved by the Voronoi tokenizer across all test datasets. While its performance is comparable to the KNN tokenizer, the Voronoi tokenizer provides a substantial boost in inference speed—over 80% faster on average—while also reducing GPU memory usage by more than 15%. As demonstrated by SAM, inference speed is critical for real-world applications, making the Voronoi tokenizer particularly valuable.

---

> > ### Comment · Reviewer_nzz5 · 2024-11-26
> >
> > Thank the authors for for their efforts during the rebuttal. The response resolved most of my concerns, especially for efficiency improvements about the Voronoi tokenizer.
> >
> > However, I wonder whether the black points in the first row of Figure 5 represents the false-positive segmentation results under 5 prompts and 10 prompts?

---

> > > ### Author Response · Authors · 2024-11-26
> > >
> > > We sincerely thank you for the invaluable feedback and positive assessment. Your expertise and the rating upgrade are truly appreciated.
> > >
> > > The black points in the first row represent the false-negative segmentation results for the ground. This issue arises from mixed segmentation labels caused by other objects lying on the ground, making it particularly challenging to accurately segment the ground, especially for points near the center of the LiDAR sensor. The ambiguity of these points adds to the difficulty, making it hard for our model to label them correctly.

---

### Official Review · Reviewer_YF5t · 2024-11-04

**Soundness:** 3
**Presentation:** 3
**Contribution:** 2
**Rating:** 6
**Confidence:** 3

**Summary:**

This paper addresses the 3D promptable segmentation task by examining it from three key aspects: task, model, and data. The authors propose a unified model named Point-SAM, which utilizes a point cloud representation and introduces a novel tokenizer based on Voronoi diagrams. In terms of data, they explore the generation of pseudo labels using SAM. The experimental results highlight the model's robust zero-shot transferability to unseen point cloud distributions and new tasks.

**Strengths:**

1. The paper presents a clear writing logic, effectively outlining the challenge to be addressed and the three distinct perspectives of the research.
2. The 3D segmentation task is a crucial direction in embodied intelligence, as it enables machines to understand and interact with complex environments. Moreover, scaling up to achieve 3D foundation models presents significant value.

**Weaknesses:**

The main consideration is the technical novelty of this paper, which leads me to feel that the overall contribution is somewhat weak.

1. In terms of model design, the method introduces the Voronoi tokenizer, which is innovative. However, compared to KNN, there is no significant performance improvement; the gains are only noticeable when the number of prompt points is low.
2. Additionally, there are many works utilizing SAM for 3D pseudo-labeling, whether in autonomous driving or in indoor settings, such as SAM3D, which also leverages multi-view consistency in 2D projections to refine results. What distinguishes the approach in this paper from previous works in terms of innovation?
3. I have some concerns about the practicality of this method. Is it can serve as a more general tool for 3D segmentation labeling? For instance, in autonomous driving, if segmentation requires placing points on every object, it seems inefficient. The challenge with object-level distinction lies in identifying the various parts, making the placement of prompt points crucial.

Besides, I would like to share some suggestions or open problems:

1. I think semantic information is crucial for 3D foundation models, as segmentation can vary in granularity, and semantics often dictate the level of segmentation.
2. Additionally, part segmentation without the need for interaction points is also valuable, such as distinguishing different parts in robotic grasping tasks.

**Questions:**

1. Could you provide more visualizations for object-level segmentation in OOD scenarios to demonstrate the generalization of the method?

2. Are there any special design considerations when training with both indoor datasets and object-level datasets together?

---

> ### Author Response · Authors · 2024-11-23
>
> > Concern about Voronoi tokenizer performance
> >
>
> We have included a more detailed Table 9 in our revision, from L813 to L824, as well as in our general response. Table 9 highlights the significant performance improvements achieved by the Voronoi tokenizer across all test datasets. While its performance is comparable to the KNN tokenizer, the Voronoi tokenizer offers a substantial boost in inference speed—over 80% faster on average—while also reducing GPU memory usage by more than 15%. As demonstrated by SAM, inference speed is crucial for real-world applications, making the Voronoi tokenizer particularly valuable.
>
> > Innovations for pseudo label generation
> >
>
> Previous lifting-based methods, such as SAM3D, use 2D SAM to segment multi-view images, with the segmentation masks across different views being associated only through semantic and 2D-3D lifting heuristics. However, these methods are not able to utilize the knowledge from existing 3D segmentation datasets, which provide richer 3D priors. By using a pretrained Point-SAM that leverages knowledge from these 3D segmentation datasets, we are able to obtain more confident pseudo-labels.
>
> > Concerns about the practicality
> >
>
> As demonstrated by SAM, promptable segmentation is highly practical, we do not concern practicality will be a critical issue. Our model also supports zero-shot instance proposals, which do not require clicking on points. We are currently working on collecting a larger training dataset that includes more diverse semantic information to further enhance the zero-shot functionality. Additionally, we are considering incorporating language prompts as part of our future work.
>
> > Utilization of semantic information
> >
>
> We agree that granularity and segmentation are closely associated with semantics; however, segmentation itself is not solely defined by semantics. Similar to SAM, our model generates multiple proposals for the first prompt point, which helps to mitigate the granularity ambiguity. Moreover, the goal of our model is to learn the process of promptable segmentation, meaning the segmentation results are determined by the prompts, rather than being heavily reliant on data biases related to granularity and semantics. We hypothesize that semantic information is already embedded within the model, as demonstrated by SAM and Point-SAM, which produce convincing segmentation results with only a few prompt points.
>
> > No interaction part segmentation
> >
>
> We agree with the reviewer that a no-interaction segmentation system is valuable for robotics tasks. In tasks such as grasping and placing, researchers commonly use 2D SAM with bounding box prompts, which are guided by bounding boxes generated by open-vocabulary detection models. However, since there is no well-known 3D open-vocabulary detection model, we aim to explore promptable segmentation with language prompts as part of our future work. The main advantage of Point-SAM comparing with SAM in downstream tasks is a 3D native segmentation model achieves much faster speed than lifting multi-view SAM results, especially for the data from reconstruction and generation model.
>
> > More visualization
> >
>
> Thank you for your suggestion. We have added additional visualization results in the revision, from L864 to L896. In this section, we present segmentation results on the Waymo segmentation dataset, which consists of real-world LiDAR scans, demonstrating the transferability of Point-SAM. We also include visualization results for few-shot probing on ShapeNet-Part, which highlights the superior pre-training quality of Point-SAM.
>
> Additionally, the videos in our supplementary material showcase the progress of promptable segmentation in complex out-of-distribution (OOD) scenarios. All scenes and objects featured in the videos are sourced either from Polycam, which provides real-world scans, or from Objaverse. We plan to include more complex examples, similar to those shown in the videos, in future versions of the paper.
>
> > Special design for training on indoor and part-level datasets
> >
>
> To train on data from these two different sources, the main design involves 1) aligning the up axis and 2) normalizing the points to the canonical space. No additional special design is required for this process.

---

> > ### Comment · Reviewer_YF5t · 2024-11-26
> >
> > Thank you to the authors for their response, which has addressed my concerns. I have raised my score accordingly.

---

> > > ### Author Response · Authors · 2024-11-26
> > >
> > > We sincerely thank you for the invaluable feedback and positive assessment. Your expertise and the rating upgrade are truly appreciated.

---

### Official Review · Reviewer_b6dA · 2024-11-04

**Soundness:** 3
**Presentation:** 3
**Contribution:** 3
**Rating:** 8
**Confidence:** 4

**Summary:**

The paper under review presents a 3D promptable segmentation model for point clouds, called Point-SAM. Similar to the architecture of Segment Anything Model(SAM), Point-SAM also contains three parts: a point-cloud encoder, a prompt encoder and a mask decoder. A Voronoi tokenizer is adopted to divide the point cloud into patch tokens. To improve the generalization of the model, multiple existing 3D segmentation datasets are included in training. Additionally, to augment part-level segmentation data, the authors utilize both pre-trained Point-SAM to obtain an initial 3D mask and then leverage SAM to refine the mask using additional views. The method demonstrates good performance on several indoor and outdoor scenes and showcases applications such as interactive 3D annotation, zero-shot 3D instance proposal and few-shot part segmentation.

**Strengths:**

1. The paper extends a powerful 2D segmentation anything model (SAM) into the 3D point cloud domain
2. It introduces a novel data engine to generate multi-level pseudo-labels and augment the training data
3. The interactive segmentation video is impressive, which illustrates its potential real-world application

**Weaknesses:**

1. Lack of Visualization Results: The authors provide only a limited number of visualizations in both the main paper and supplementary materials. The qualitative results presented in Figure 4 are inadequate. It is anticipated that more complex examples, such as those depicted in the supplementary videos, could be included. Moreover, several applications discussed in the paper, such as few-shot part segmentation and zero-shot object proposal generation, lack corresponding visualization results.

2. In line 40, the paper asserts that "multi-view images only capture the surface, making it infeasible to label internal structures." However, the paper does not include examples, such as drawers, to demonstrate the superior performance of Point-SAM in these scenarios. The authors are expected incorporate additional experiments or examples to make this claim more convincing.

3. The segmentation quality does not look good: The paper does not directly compare its results with those of other 3D frameworks that merge results across multiple views, despitethe experiments on MVSAM. While it is acknowledged that most of these frameworks require post-processing, which can be time-consuming, their segmentation quality reported in those papers are superior, particularly along boundary regions, such as teh results in SAM3D and more advanced methods like Gaussian Grouping.

4. Prompt Point Selection: Could the authors provide a more detailed explanation of how point prompts are selected in the experiments? In Figure 4, the points are not consistently placed, and some points are marked as negative in one method while none are negative in another. Also. the positions of points are different.

**Questions:**

Please refer to the "weaknesses" section

---

> ### Author Response · Authors · 2024-11-23
>
> > Lack of visualization
> >
>
>
> Thank you for pointing out the lack visualization in our paper. We have added more qualitative results in the appendix, from L864 to L896. In this section, we demonstrate the transferability of our model by applying it to the Waymo segmentation dataset, which is an out-of-distribution (OOD) outdoor LiDAR dataset. We also show the few-shot segmentation results comparing with Uni3D, which indicates the superior pre-training quality of our model.
>
> We also thank your suggestion regarding adding more complex examples, and we appreciate your interest in the videos in the supplementary material, which demonstrate the superior transferability of our model. We will include additional visualization examples in our next revision to further illustrate the model's capabilities.
>
> > Evaluation on data with interior points
> >
>
> We want to clarify that the point clouds in our original PartNet-Mobility evaluation set were generated by lifting multi-view RGBD images, ensuring a fair comparison with MV-SAM. This demonstrates that Point-SAM already outperforms baseline methods on data containing only surface points, whereas multi-view lifting-based methods are incapable of handling interior points.
>
> To further substantiate our claim, we conducted an experiment using the StorageFurniture category from PartNet-Mobility, which includes interior segmentation masks. The results have been added to the revised manuscript as Table 11. In this experiment, Point-SAM was trained on PartNet, ShapeNet, and ScanNet, excluding PartNet-Mobility data from the training set.
>
> As shown in Table 11, MV-SAM performs worse compared to its results in Table 3. Point-SAM also shows slightly lower performance compared to the results in Table 4. This drop is attributed to the fact that ShapeNet, a part of Point-SAM's training data, contains point clouds derived from multi-view images. Despite this, Point-SAM delivers strong performance and still surpasses MV-SAM.
>
> > Comparison with lifting methods
> >
>
> To the best of our knowledge, lifting-based methods such as SAM3D, SAMPro3D, and Gaussian Grouping primarily focus on zero-shot instance grouping and do not support promptable segmentation with interactively selected prompt points. Furthermore, these methods are limited to point clouds with associated multi-view images and camera parameters. They are also inapplicable to datasets with interior points. To address the challenge of interactive promptable segmentation for aggregation-based methods, we introduce MV-SAM, which combines the interactively prompted segmentation results from all views. For each view, we separately prompt SAM using the prompt points sampled specifically on that view.
>
> In contrast, our method is more versatile, as it can be applied to any point cloud, regardless of its source or the presence of multi-view images or interior points. We are glad to hear your suggestions on how to adapt those lifting-based methods for 3D promptable segmentation. Note that those prior works usually rely on global hyperparameters like IoU thresholds to merge instances across views, which makes it difficult to improve performance even with users’ prompts.
>
> > Prompt points selection
> >
>
> We follow the prompt point selection rule of AGILE3D, as described in Appendix C, which is also used in works on 2D promptable segmentation. The first point is selected at the center of the ground truth mask. For subsequent iterations, two prompt candidates are selected from the centers of false positive and false negative points. The candidate with the larger distance to the border of its region is then chosen as the prompt point.
>
> In our experiments, we follow this same rule to select prompt points for both our method and the baselines. Therefore, the first prompt point is identical for all methods, but the subsequent prompt points differ, depending on the output of each method. Both the positions and labels of subsequent prompt points are determined by the method's output and the selection rule, which results in differences between our method and the baselines.

---

> > ### Comment · Reviewer_b6dA · 2024-11-26
> >
> > Thanks for the authors' reply. I do not have major concerns except the segmentation quality demonstrated in the videos, which is not very accurate. However, I think it is acceptable.
> >
> > Here are some minor questions:
> > 1. The proposed data engine leverages SAM and iteratively obtain the 3D masks by projection. Accurately, this idea has been adopted in some existing methods such as SA3D [1]. I suspect that the final mask might degrade to some unexpected mask if any intermedian step is incorrect during this iterative process. Does this problem happen in your experiment? If so, does it effect the training?
> > 2. In Table 5,  PointSAM* is trained on ScanNet following the setting of AGILE3D. So I think that the data engine is not included, is it? However, the performance on KITTI360 is extremely good, which looks strange since the model is trained on ScanNet only and KITTI360 is totally out of distribution from my perspective. Can the author explain more on this?
> >
> >
> >
> > [1] Cen, J., Fang, J., Zhou, Z., Yang, C., Xie, L., Zhang, X., Shen, W. and Tian, Q., 2023. Segment Anything in 3D with Radiance Fields. arXiv preprint arXiv:2304.12308.

---

> ### Author Response · Authors · 2024-11-26
>
> We sincerely thank the reviewer for the invaluable response.
>
> > Concern about segmentation quality demonstrated in videos
> >
>
> The segmentation results in the video were displayed on mesh faces, and some inaccuracies arose from the projection of point clouds onto the corresponding meshes. Additionally, these inaccuracies were also due to the insufficient number of prompt points, which limited the accuracy of the segmentation results on those totally out of distribution objects.
>
> To address your primary concern regarding the segmentation quality shown in our video, we have included additional visualization results in Figure 7 of our latest revision. As demonstrated in Figure 7, when provided with sufficient prompt points, our model effectively handles complex objects and scenes sourced from Polycam and Objaverse websites. This underscores Point-SAM's robustness in processing entirely out-of-distribution data. We hope these enhanced visualizations can address your concerns.
>
> >Pseudo mask degrade during iterative refinement
> >
>
> Yes, we observed this phenomenon during the development of our data engine, particularly while working on the first version, which was inspired by SAMPro3D. During this phase, we focused on leveraging 3D space consistency and heuristics, and we observed that the small errors in some iterations will be accumulated and finally make the result mask degrade to another unexpected mask. To mitigate this issue, we designed our data engine with Point-SAM as the bridge of the views, which provides more accurate masks on other views.
>
> We qualitatively compared the segmentation results between the data engine with and without Point-SAM and found that the pre-trained Point-SAM, trained on datasets with segmentation labels, significantly helped to mitigate this issue, although some failure cases still remain. We then compared the Point-SAM results with the pseudo labels generated from both data engines and evaluated them on the PartNet validation dataset. The results are shown below. Based on these findings, we believe that mask degradation does impact the training, but the Point-SAM-based data engine helps alleviate this phenomenon.
>
> | data engine | PartNet Validation |
> | --- | --- |
> | SAMPro3D | 78.6 |
> | w/ Point-SAM | **80.5** |
>
> > Good performance of Point-SAM trained on ScanNet
> >
>
> We believe the good performance on KITTI360 is due to the relatively easier segmentation challenges presented by this dataset. Compared to datasets with part annotations, the instances in KITTI360 are typically spatially separated, a characteristic that is also observed in ScanNet. Additionally, the lack of contact between foreground instances further simplifies the segmentation task. Point-SAM performs well because it learns an iterative, promptable segmentation program on ScanNet, refining results progressively based on prompts. Since Point-SAM is designed for transferring across different datasets—such as using normalized point clouds in a canonical space—the performance on KITTI360 is both expected and reasonable. On the other hand, we believe the relatively lower performance of AGILE3D on KITTI360 is due to the fact that AGILE3D has more hyper-parameters that require fine-tuning when transferring to a new dataset. For example,  the point clouds are not normalized and the SparseConv backbone is highly sensitive to scale.

---

### Official Review · Reviewer_AfEp · 2024-11-04

**Soundness:** 3
**Presentation:** 3
**Contribution:** 3
**Rating:** 8
**Confidence:** 4

**Summary:**

This paper addresses the problem of deploying 3D foundational models by introducing a novel 3D promptable segmentation model for point clouds, named Point-SAM. This model extends the Segment Anything Model (SAM) into the 3D domain by developing a promptable 3D architecture that utilizes both point and mask prompts. These prompts, together with the input point cloud, are processed through a transformer-based encoder-decoder architecture to generate a 3D segmentation mask.

The Point-SAM model first tokenizes the input point cloud using one of two approaches: a single set abstraction layer (based on PointNet++), which creates patch-based features centered at each patch’s centroid encoding local geometry, or a Voronoi partition-based tokenizer, where  each Voronoi cell represents the patch’s centroid, and point-wise features are max-pooled  within each cell. The Voronoi-based method reduces computational and memory costs by bypassing the need for dense point-wise feature extraction in each centroid's vicinity. Patch features are then extracted through a ViT model, and mask prompts are incorporated from previous decoder iterations. The model upscales these patch embeddings to the original point cloud resolution using 3-nn inverse distance weighting. Following this, a two-way transformer facilitates interaction between point prompts and point cloud embeddings by utilizing cross-attention operations. The final 3D proposal is derived by applying an output token processed through a MLP.

Point-SAM is trained on a diverse set of datasets that include part-level and object-level annotations across single-object and scene-level modalities. To improve generalizability on out-of-distribution data, the ShapeNet dataset is used to transfer knowledge from SAM. This is achieved by generating 2D pseudo masks that serve as prompts to a pretrained Point-SAM model, iteratively refining its 3D proposals.

Point-SAM's is evaluated on several benchmarks for the tasks of zero-shot point-prompted segmentation, zero-shot object proposal and few-shot part segmentation, where it achieves performance that is either on par with or surpasses competing methods.

**Strengths:**

+ The method is a step forward towards 3D foundational models, eliminating the need of using multiple 2D views of the 3D object and 2D-3D lifting of SAM proposals at inference, while it provides the ability of refining the 3D proposals with additional prompts (as seen in supp.).
+ Unified training strategy on 3D point clouds across several datasets, covering different modalities either at annotation or scale level (part, object masks; single object, entire scenes).
+ Good performance on zero-shot point-prompted segmentation w.r.t. alternative methods.
+ Knowledge distillation from 2D foundational models such as SAM during training to further enhance the generalizability of Point-SAM.

**Weaknesses:**

- The proposed Voronoi diagram tokenizer while it manages to lower the computational and memory cost of the overall pipeline, in many cases it fails to surpass the performance of the k-nn based tokenizer.
- Regarding the OOD scenarios and particularly the PartNet-Mobility, the held-out categories (scissors, refrigerators, and doors) are all part of the PartNet, thus Point-SAM has seen these during training. This weakens the zero-shot transferability of the method.
- The process of generating pseudo labels and transferring knowledge from SAM is somewhat unclear (see Questions).

**Questions:**

- Regarding the generation of pseudo labels, lines 265-269 state that the generated 3D proposal, using the 1st view’s random selected 2D prompt from SAM’s proposal, is projected back into the same view. This is then used to sample a 2D prompt (negative prompt) from the error region between the two (2D-3D proposal). Based on Figure 3, this is done in the next view (view 2). The first is used to sample the 2D prompt, the generated 3D proposal is used to prompt SAM with the 2nd view as input, and then the negative prompt is sampled for the generated 2nd 2D proposal. So, which of the two is the correct procedure?
- In Table 3, the MV-SAM baseline is absent for the ScanObjectNN dataset. Additionally, it's unclear which specific approach the term “InterObject3D++” refers to (lines 391 and 395). Is this the same baseline used in AGILE3D?
- In Figure 4, what is the difference of blue and red point prompts?

---

> ### Author Response · Authors · 2024-11-23
>
> > Voronoi vs. KNN
> >
>
> Thank you for pointing out the slightly lower performance of the Voronoi diagram tokenizer. We have added a table in the revision (L827 - L844) and the general response above to provide additional evidence supporting our claim in L377.
>
> We would like to emphasize that the inference speed is critical to the downstream tasks, especially like 3D interactive annotation. The Voronoi tokenizer significantly enhances inference speed by over 80% on average and reduces GPU memory usage by more than 15%. Note that our Voronoi tokenizer performs on par with the KNN tokenizer (within 4% IoU in Table 3), and much better than the baselines.
>
> > OOD evaluation categories in training data
> >
>
> To conduct the OOD evaluation, we also excluded the point clouds of these three categories from the PartNet dataset (L248-L249).
>
> > The process of generating pseudo labels
> >
>
> Thank you for pointing out the confusion. The description of pseudo-label generation in our paper (L262 - L302) provides a detailed explanation of our method. For simplicity in Figure 3, we omitted the refinement process for the first view and illustrated the same procedure for the other views.
>
> We perform 2D-3D refinement on both the first view and the other views. For the first view, the refinement is used to correct the Point-SAM proposal, ensuring alignment with the initial 2D mask while keeping some inductive biases from SAM (since the refinement from SAM is not perfect). For the other views, refinement is applied after obtaining the 2D SAM mask proposal by projecting prompt points from the 3D proposed mask. This process leverages the prior knowledge from SAM to perform spacing carving on the other views.
>
> > MV-SAM baseline not applied to ScanObjectNN
> >
>
> Lifting-based methods like MV-SAM rely heavily on high-quality 2D renderings. Since ScanObjectNN provides only sparse point clouds without associated multi-view images, obtaining high-quality 2D renderings is challenging. Therefore, we compare with MV-SAM only on PartNet-Mobility, which provides textured meshes. Note that our MV-SAM is modified from SAM and SAMPro3D for 3D promptable segmentaiton, while SAMPro3D or other similar works are intended to generate instance segmentation proposals.
>
> InterObject3D++ is the same baseline mentioned by AGILE3D.
>
> > Legends in Figure 4
> >
>
> Thank you for pointing it out. In Figure 4, the red points represent positive prompt points, and the blue points stand for negative prompt points. We have updated the caption.

---

> > ### Comment · Reviewer_AfEp · 2024-11-26
> >
> > I thank the authors for their response. My main issue was the Voronoi tokenizer. While it shows minor IoU performance drops, it significantly outperforms the k-NN variant in speed and memory efficiency. I will increase my score accordingly.

---

> ### Author Response · Authors · 2024-11-26
>
> We sincerely thank you for the invaluable feedback and positive assessment. Your expertise and the rating upgrade are truly appreciated.

---

### Author Response · Authors · 2024-11-23

We sincerely appreciate the reviewers for dedicating their time and effort to reviewing our work. We are also grateful for their recognition of our contributions to advancing 3D promptable segmentation. Below, we address each reviewer's concerns and have updated the submission based on their valuable suggestions.

One common question raised by multiple reviewers concerns the performance analysis of the Voronoi-based tokenizer. The main intuition behind this tokenizer is to enhance the efficiency of Point-SAM, thereby improving its practicality. While it achieves comparable effectiveness to the KNN tokenizer, its superior efficiency highlights the importance of the Voronoi tokenizer, which is crucial for real-world applications. The results are presented in Table 9 of our revised submission, and we have also included the table below for clarity.

| Datasets | Method | FPS@1 | FPS@5 | FPS@10 | Memory |
| --- | --- | --- | --- | --- | --- |
| PartNet-Mobility | KNN | 23.2 | 11.7 | 7.3 | 2524 |
|  | Voronoi | **28.6 (+34.4%**) | **22.4 (+82.0%)** | **17.8 (+143.8%)** | **2078 (-21.5%)** |
| ScanObjectNN | KNN | 5.9 | 3.0 | 1.4 | 8548 |
|  | Voronoi | **7.3 (+23.7%)** | **4.9 (+63.3%)** | **2.3 (+62.0%)** | **6316 (-26.1%)** |
| S3DIS | KNN | 3.31 | 1.57 | 0.93 | 13680 |
|  | Voronoi | **3.52 (+34.4%)** | **1.72 (+8.7%)** | **1.04 (+11.8%)** | **10788 (-20.5%)** |
| KITTI360 | KNN | 15.7 | 8.9 | 5.2 | 3890 |
|  | Voronoi | **21.1 (+34.4%)** | **16.2 (+82.0%)** | **13.7 (+163.4%)**  | **3172 (-18.4%)** |

---

### Meta-Review · Area_Chair_doaA · 2024-12-17

**Metareview:**

This paper receives all positive ratings of 8,8,6,6,6,6. The AC follows the recommendation of the reviewers to accept the paper. All reviewers think that the work on 3D promptable segmenetation model of Point-SAM is useful for the community and the proposed method is good. The weaknesses mentioned by the reviewers are mainly clarifications and are well-addressed in the rebuttal and discussion phases.

**Additional Comments On Reviewer Discussion:**

The weaknesses mentioned by the reviewers are mainly clarifications and are well-addressed in the rebuttal and discussion phases.

---

### Decision · Program_Chairs · 2025-01-22

Accept (Poster)